# Text-promptable Propagation for Referring Medical Image Sequence Segmentation

## Abstract

Medical image sequences, generated by both 2D video-based examinations and 3D imaging techniques, consist of sequential frames or slices that capture the same anatomical entities (e.g., organs or lesions) from multiple perspectives. Existing segmentation studies typically process medical images using either 2D or 3D methods in isolation, often overlooking the inherent consistencies among these images. Additionally, interactive segmentation, while highly beneficial in clinical scenarios, faces the challenge of integrating text prompts effectively across multi-modalities. To address these issues, we introduce an innovative task, **Referring Medical Image Sequence Segmentation** for the first time, which aims to segment the referred anatomical entities corresponding to medical text prompts. We develop a strong baseline model, Text-Promptable Propagation (TPP), designed to exploit the intrinsic relationships among sequential images and their associated textual descriptions. TPP supports the segmentation of arbitrary objects of interest based on cross-modal prompt fusion. Carefully designed medical prompts are fused and employed as queries to guide image sequence segmentation through triple-propagation. We curate a large and comprehensive benchmark covering 4 modalities and 20 different organs and lesions. Experimental results consistently demonstrate the superior performance of our approach compared to previous methods across these datasets. Code and data are available at TPP.

## 1 Introduction

In the realm of medical imaging, both 2D video-based examinations, such as endoscopy and ultrasound, and 3D imaging techniques like CT and MRI, produce sequential frames or slices that capture the same anatomical entities (e.g., organs and lesions). These image sequences are not merely collections of individual snapshots but are deeply interconnected, with each frame or slice providing a unique view of the same object from different angles and in varying shapes. The consistencies among these sequential images are crucial for comprehensive medical analysis and diagnosis.

Current studies on medical image segmentation involve advanced machine learning tools designed to automatically identify and separate different organs, tissues, or pathological regions from medical images like CT scans, MRIs, and X-rays. These medical segmentation models are essential for tasks such as disease screening, organ segmentation, and anomaly detection. As depicted in Figure 1 (a-c), these models face two primary drawbacks: 1) Separate network architectures for 2D and 3D medical images. Researchers often apply distinct methodologies for 2D and 3D medical images, using 2D approaches for planar images or slices (Ronneberger et al., 2015; Chen et al., 2021) and 3D techniques for volumetric data (Milletari et al., 2016; Zhao et al., 2023). These networks neglect the consistency across temporal frames from 2D video-based examinations and sequential slices from 3D volumes when images are interrelated or part of a sequence, leading to discrepancies and sub-optimal outcomes. 2) Limitation to closed sets of categories and lack of human interactions. In multi-class segmentation tasks, existing methods (Chen et al., 2021; Zhao et al., 2022) typically restrict results to predefined classes, reducing flexibility and preventing the specification of particular classes for referring segmentation. This rigidity limits the adaptation of segmentation processes to specific clinical needs or emerging requirements in complex medical scenarios.

To address these challenges, we introduce an innovative task, **Referring Medical Image Sequence Segmentation**, which aims to identify and segment anatomical entities corresponding to given text

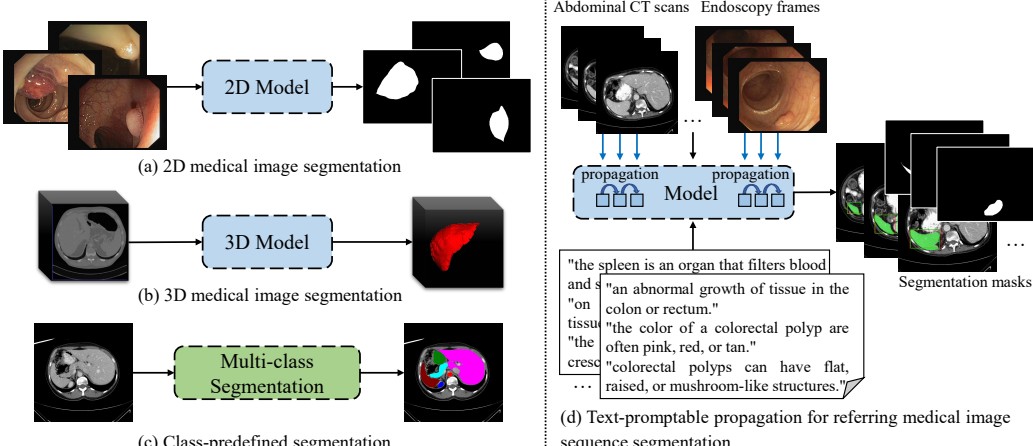

Figure 1: Medical image segmentation. (a) 2D models are often applied to 2D images or slices from 3D volumes, (b) 3D images typically utilize 3D models, (c) in multi-class segmentation tasks, once the categories are defined, the prediction results are restricted to those categories, without the ability to specify a specific category to segment, and (d) our method leverages medical text prompts to refer to target objects and treats frames from 2D video-based examinations and slices from 3D volumes as medical image sequences for segmentation. This approach enhances the flexibility and accuracy by integrating text-based references with a broader range of image data.

prompts within medical image sequences. These sequences involve both temporally related frames from videos and spatially related slices in volumes. For this task, we present our **T**ext-**P**romptable **P**ropagation (TPP) model, a strong baseline designed to leverage the intrinsic relationships among sequential images and their associated textual descriptions. As shown in Figure 1 (d), TPP unifies frames from 2D video-based examinations and slices from 3D volumes, and supports the segmentation of arbitrary objects of interest based on text prompts.

TPP integrates two key components: 1) **Cross-modal Prompt Fusion.** This component supports medical text prompts and vision-language fusion. Medical text prompts often provide valuable knowledge and context by highlighting specific regions of interest and guiding attention. We propose cross-modal prompt fusion to integrate prompts that describe various characteristics of anatomical entities, linking medical image sequences with text prompts across vision and language modalities. This component facilitates a more comprehensive understanding of the data by combining insights from both textual and visual information. 2) Transformer-based model with **Triple Propagation**. To uniformly model the temporal relationships between 2D frames and cross-slice interactions in 3D volumes, we employ a Transformer-based encoder-decoder architecture that incorporates propagation strategies to track the referred objects throughout the sequences.

We curate a large dataset for referring medical image sequence segmentation, Ref-MISS, by prompting Large Language Models and re-organizing public medical datasets. Ref-MISS is sourced from 18 diverse medical datasets across 4 modalities, including MRI, CT, ultrasound, and endoscopy. It covers 20 different organs and lesions from various regions of the body, as illustrated in Figure 2.

## 2 RELATED WORK

**Medical Image Segmentation.** As mentioned earlier, researchers typically apply distinct methods for 2D (Ronneberger et al., 2015) and 3D (Çiçek et al., 2016; Milletari et al., 2016) medical images. 2D models are used for planar images or slices, while 3D models are intended to learn volumetric features. Isensee et al. (2021) introduced a versatile, self-adaptive deep learning framework specifically designed for medical image segmentation tasks, extending the U-Net architecture and its 3D version. Chen et al. (2021) pioneered the combination of Transformer-based architecture with Convolutional Neural Networks (CNNs) for medical image segmentation, applying a slice-by-slice inference on 3D volumes without considering interrelationships among slices. Some works (Ji et al.,

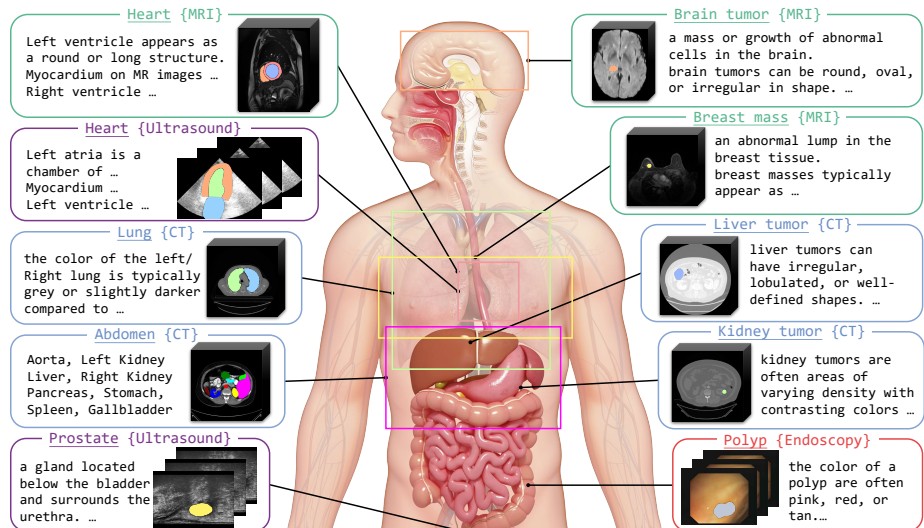

Figure 2: An illustration of focus areas in Ref-MISS dataset.

2021; Painchaud et al., 2022; Lin et al., 2023) utilize spatial-temporal cues to enhance segmentation performance; however, these models are limited to specific modalities and tasks.

**Medical Vision Language Models.** Medical vision language models have achieved success across multiple downstream tasks, including diagnosis classification (Moon et al., 2022; Wang et al., 2022; Tiu et al., 2022; Lu et al., 2023), lesion detection (Qin et al., 2023; Huang et al., 2024), image segmentation (Zhao et al., 2023; Li et al., 2023), report generation (Yan & Pei, 2022; Bannur et al., 2023), and visual question answering (Singhal et al., 2023; Moor et al., 2023). Qin et al. (2023) designed auto-generation strategies for medical prompts and transferred large vision language models for medical lesion detection. Zhao et al. (2023) built a model based on Segment Anything Model (Kirillov et al., 2023) in medical scenarios driven by text prompts, but the model focused on 3D medical volume segmentation and failed to account for the sequential relationships between scans. Based on our current understanding, we are the first to treat 2D and 3D medical images as unified medical image sequences, using medical text prompts to specify segmentation targets.

**Referring Video Object Segmentation.** Gavrilyuk et al. (2018) were the first to propose inferring segmentation from a natural language input, extending two popular actor and action datasets with natural language descriptions. Seo et al. (2020) constructed the first large-scale referring video object segmentation (RVOS) dataset and proposed a unified referring video object segmentation network. Wu et al. (2022) and Botach et al. (2022) presented Transformer-based RVOS frameworks, enabling end-to-end segmentation of the referred object. Wu et al. (2023) designed explicit query propagation for an online model. Luo et al. (2024) aggregated inter- and intra-frame information via a semantic integrated module and introduced a visual-linguistic contrastive loss to apply semantic supervision on video-level object representations. Inspired by these works, we introduce a new task termed Referring Medical Image Sequence Segmentation. We process both 2D and 3D medical data into image sequences and perform an in-depth exploration of clip-level consistency within the sequences under the guidance of linguistic prompts.

## 3 METHOD

Given $T$ frames or slices $\{I_t \in \mathbb{R}^{3 \times H \times W}\}_{t=1}^{T}$ from a medical image sequence clip and $N_p$ medical text prompts $\{P_i\}_{i=1}^{N_p}$, the model aims to predict the segmentation masks $\{\hat{m}_t \in \mathbb{R}^{H \times W}\}_{t=1}^{T}$ of the referred object corresponding to the prompts. We provide an automatic text-promptable schema for referring medial image sequence segmentation. The overall architecture of our framework is illustrated in Figure 3 (a). This framework comprises vision-language fusion (Section 3.1), unified sequential processing (Section 3.2), and the training and inference procedures (Section 3.3).

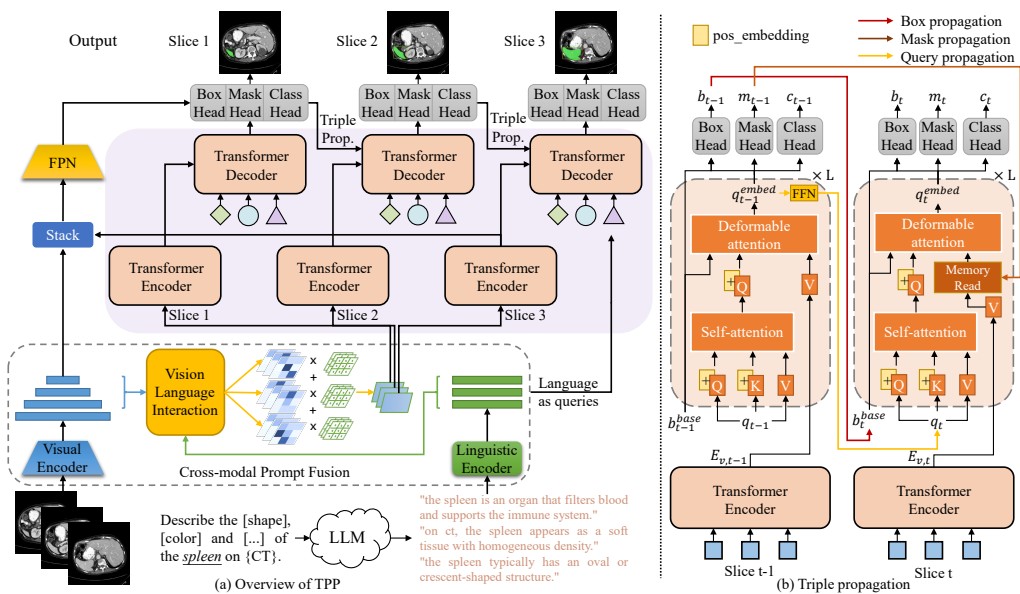

Figure 3: (a) Overall architecture of our Text-Promptable Propagation for referring medical image sequence segmentation. Triple Prop. is short for Triple Propagation. (b) Illustration of Triple Propagation in Transformer decoder, consisting of box-level, mask-level, and query-level propagation. Line from $E_{v,t-1}$ to Memory Read block is omitted for simplicity.

## 3.1 VISION-LANGUAGE FUSION

**Prompt Acquisition.** We adopt large language models to automatically generate medical text prompts. The instruction template is as follows: "You are a medical expert. Describe the [attribute 1], [attribute 2], ..., and [attribute $N_p$] of the *anatomical entity* on {modality} in one sentence each." Using this template, we obtain $N_p$ prompts for the target object (i.e., anatomical entity) that is expected to be segmented. Here, $N_p$ is set to 3, with [attribute 1]=[profile], [attribute 2]=[shape], and [attribute 3]=[color]. The attribute [profile] conveys the characterization of organ functions and the definition of lesions, while attributes [color] and [shape] provide detailed descriptions of the morphological aspects of the object.

**Feature Extraction.** Image clips $\{I_t\}_{t=1}^T$ and medical text prompts $\{P_i\}_{i=1}^{N_p}$ are fed into a visual encoder and a linguistic encoder separately to extract visual features $F_v$ for each image and textual features $F_p$ for each prompt. $F_v$ is a set of feature maps $\{f_v^l \in \mathbb{R}^{C^l \times H^l \times W^l}\}_{l=1}^4$, where $C^l$, $H^l$ and $W^l$ denote the channel dimension, height, and width of the feature map at the $l^{th}$ level, respectively. $F_p$ is a set of word-level embeddings $\{f_p^i \in \mathbb{R}^{len_i \times C}\}_{i=1}^{N_p}$, where $len_i$ and $C$ denote the sentence length and channel dimension of the $i^{th}$ prompt, respectively.

**Cross-modal Prompt Fusion.** Having obtained the visual and textual features, we proceed with Cross-modal Prompt Fusion. This module enhances the focus on target objects within images by leveraging text prompts and assists in selecting the most relevant and useful prompt for each specific clip. The process involves three key steps. 1) For features of each image, we first apply Multi-Head Attention (MHA) mechanisms between the visual feature maps at the last three levels and the word-level embeddings from the text prompts. Each text prompt leads to corresponding proposals, denoted as $\mathbb{A}, \mathbb{B}, \mathbb{C}$, respectively. This allows us to capture intricate relationships between the visual and textual data. 2) Our goal is to identify the target object, i.e. $\mathbb{A} \cap \mathbb{B} \cap \mathbb{C}$. The attention output is then flattened and passed through a three-layer Multi-Layer Perceptron (MLP) to compute weights for each text prompt. These weights reflect the relevance of each prompt to the current clip. Using the computed weights, we perform a weighted sum of the attention output to obtain the fused visual features. This step integrates the most pertinent aspects of the text prompts with the visual data. The process can be formulated as:

$$A^{l,i} = \text{MHA}\left(f_v^l, f_p^i\right), \tag{1}$$

$$W^{l,i} = \text{Softmax}\left(\text{MLP}\left(A^{l,i}\right)\right), \tag{2}$$

$$F_v' = \left\{\sum_{i=1}^{N_p} f_v^l \cdot A^{l,i} \cdot W^{l,i}\right\}_{l=2}^{4}. \tag{3}$$

3) For the textual features, we select the prompt with the highest weight score produced by the feature maps at the first level ($l = \{1\}$). The textual feature of this prompt is then fed into the Transformer decoder as the query input.

$$\hat{w} = \underset{i \in \{1,\dots,N_p\}}{\arg\max}\left(W^{l=1,i}\right), \tag{4}$$

$$F_p' = f_p^{\hat{w}}. \tag{5}$$

### 3.2 Unified sequential processing

**Transformer.** Our Transformer architecture is adapted from Deformable DETR (Zhu et al., 2021). For each image $I_t$, the Transformer encoder takes the flattened visual features $F_{v,t}'$ and 2D positional encoding as input, producing encoded output $E_{v,t}$ through multi-scale deformable attention and several feed forward layers. The output of the Transformer encoder $E_{v,t}$ and the textual feature of the selected prompt $F_{p,t}'$ are then fed into the Transformer decoder. We repeat $F_{p,t}'$ $N_q$ times to introduce $N_q$ queries, denoted as $q_t$. Meanwhile, each image receives sequential cues from the previous frame (except for the first image) in temporal order. The Transformer decoder thus generates $N_q$ embeddings for each image, denoted as $q_t^{embed}$.

**Prediction Heads.** Three prediction heads are constructed following the Transformer decoder. The output embeddings from the Transformer decoder, $q_t^{embed}$, are then processed by these prediction heads. 1) The **box head** consists of a three-layer feed-forward network (FFN) with ReLU activation, except for the last layer, which predicts the box offset. The offset is added to the base box coordinates to determine the location of the referred object, denoted as $b_t$. 2) The **mask head** is implemented by dynamic convolution (Tian et al., 2020). It takes multi-scale features from the feature pyramid network (FPN) $f_m$, concatenates them with relative coordinates, and uses a controller to generate convolutional parameters $\theta_t$. Conditional convolution is then applied to the visual features to generate $N_q$ segmentation masks $m_t$.

$$\theta_t = \text{Controller}\left(q_t^{embed}\right), \tag{6}$$

$$\left\{m_t^i\right\}_{i=1}^{N_q} = \left\{\phi^i\left(f_m; \theta_t^i\right)\right\}_{i=1}^{N_q}. \tag{7}$$

Here, the controller is also a three-layer FFN with ReLU activation, except for the last layer, and $\phi^i$ represents three $1 \times 1$ convolutional layers with 8 channels per query, using parameters $\theta_t^i$ generated by the controller. 3) Since our text prompts contain class information, the **class head** indicates whether the object is referred by the text prompt.

**Triple Propagation.** Frames or slices in temporal order across a sequence of medical images often exhibit consistency in appearance or spatial relationships. To take advantage of this temporal coherence, we propagate the box, mask, and query embeddings derived from the previous image to assist in the prediction for the current image, as depicted in Figure 3 (b). This triple propagation leverages the temporal consistency within the sequence, enhancing the robustness and precision of medical image sequence analysis and ultimately contributing to more reliable outcomes.

Given the outputs of the previous image $y_{t-1} = \{b_{t-1}^i, m_{t-1}^i, c_{t-1}^i\}_{i=1}^{N_q}$, we first choose the prediction with highest class score as the best prediction: $\{b_{t-1}^{\hat{n}}, m_{t-1}^{\hat{n}}, c_{t-1}^{\hat{n}}\}$. Consequently, except for the first image, which has $N_q$ queries, subsequent images will contain only one query, as it is always propagated from the best prediction of the previous image.

***Box-level Propagation.*** The box coordinates from the previous image $b_{t-1}^{\hat{n}}$ provide a valuable reference for estimating the location of the target object in the current image. We use these coordinates as the initial box for the current image, i.e. $b_t^{base}$, leveraging the spatial continuity between images to more accurately predict the object's position. Box-level propagation allows us to refine the object's localization by starting from a well-informed estimate.

***Mask-level Propagation.*** Similarly, the visual features encoded by the Transformer encoder $E_{v,t-1}$ and the segmentation mask $m_{t-1}^{\hat{n}}$ from the previous image contains essential semantic information that can significantly aid in analyzing the current image. To effectively utilize this prior knowledge, we employ a memory-read mechanism inspired by Space-Time Memory (Oh et al., 2019). The difference is that we only generate key and value maps for the memory. The memory map $M_{t-1}$ is a concatenation of $m_{t-1}^{\hat{n}}$ and the first level of $E_{v-1,t}$, and the memory read operation is defined as:

$$M_{t-1} = \text{Concat}\left(m_{t-1}^{\hat{n}}, E_{v,t-1}^{l=2}\right), \tag{8}$$

$$K = \psi\left(M_{t-1}\right), V = \varphi\left(M_{t-1}\right), \tag{9}$$

$$E_{v,t}^{l=2} = \text{Softmax}\left(\frac{E_{v,t}^{l=2} K}{\sqrt{C^{l=2}}}\right) V, \tag{10}$$

where $\psi$ and $\varphi$ are two parallel $3 \times 3$ convolutional layers. The first level of $E_{v,t}$ is now a memory-read map. It is concatenated with feature maps of other levels and then fed into the deformable attention module in the Transformer decoder after flattening.

***Query-level Propagation.*** Having confirmed the query index $\hat{n}$, we propagate the corresponding output query embedding $q_{t-1}^{embed}$ to the current image. Here, we use a three-layer FFN to transform the embedding to $q_t$, following (Wu et al., 2023). The propagation of the query allows for the transmission of embedded context for the same target.

### 3.3 TRAINING AND INFERENCE

**Training.** For each image, we have $N_q$ predictions $y_t = \{b_t^i, m_t^i, c_t^i\}_{i=1}^{N_q}$, where $b_t^i \in \mathbb{R}^4$, $m_t^i \in \mathbb{R}^{\frac{H}{4} \times \frac{W}{4}}$, and $c_t^i \in \mathbb{R}^1$ represent the box location, segmentation mask, and probability of the referred object, respectively. The ground-truth, in the same format, is denoted as $Y_t = \{B_t, M_t, C_t\}$. We then compute the matching loss $\mathcal{L}_{match}$ to find the best prediction:

$$\mathcal{L}_{match,t}\left(y_t, Y_t\right) = \lambda_{box}\mathcal{L}_{box}\left(y_t, Y_t\right) + \lambda_{mask}\mathcal{L}_{mask}\left(y_t, Y_t\right) + \lambda_{cls}\mathcal{L}_{cls}\left(y_t, Y_t\right), \tag{11}$$

$$\hat{n}_{q,t} = \underset{i \in \{1,...,N_q\}}{\arg\min}\left(\mathcal{L}_{match,t}\right), \tag{12}$$

where $\lambda_{box}$, $\lambda_{mask}$, and $\lambda_{cls}$ are loss coefficients. $\mathcal{L}_{box}$ is implemented as a sum of L1 loss and GIoU loss, $\mathcal{L}_{mask}$ combines Dice loss and binary mask focal loss, and $\mathcal{L}_{cls}$ is focal loss. $\hat{n}_{q,t}$ represents the query index of the best prediction. The network is optimized by minimizing the summation of $\mathcal{L}_{match,t}$ for the best predictions across $T$ images.

$$\mathcal{L} = \frac{1}{T} \sum_{t=1}^{T} \mathcal{L}_{match,t}^{\hat{n}_{q,t}}. \tag{13}$$

**Inference.** During inference, we select the query with the highest class score as the best prediction, which can be formulated as:

$$\hat{n}'_{q,t} = \underset{i \in \{1,...,N_q\}}{\arg\max}\left(c_t^i\right). \tag{14}$$

The final segmentation masks for each image $\{\hat{m}_t\}_{t=1}^{T}$ are selected using the query index $\hat{n}'_{q,t}$ from the $N_q$ predictions $\{m_t^i\}_{i=1}^{N_q}$. Due to our propagation strategy, the best prediction of the first image is propagated to the subsequent images, leading to only one query for the rest images. Therefore, for those images with only one query, $\hat{m}_t = m_t (t > 1)$.

## 4 EXPERIMENTS

### 4.1 DATASETS AND METRICS

**Datasets and Metrics.** We have collected and processed 18 medical image sequence datasets, including 20 anatomical entities across 4 different imaging modalities, as shown in Figure 2 and Table 1. The datasets are categorized by the 4 modalities as follows: 1) MRI datasets. 2018 Atria Segmentation Data (Xiong et al., 2021), RVSC (Petitjean et al., 2015), ACDC (Bernard et al.,

Table 1: Medical image sequence datasets across 4 modalities and 20 anatomical entities.

| Dataset | Class | Type | Modality | Training cases (images) | Testing cases (images) |
|---------|-------|------|----------|-------------------------|------------------------|
| Xiong et al. (2021) | Left atrium | Organ | MRI | 100 (5817) | 54 (3202) |
| RVSC | Right ventricle | Organ | MRI | 16 (243) | 32 (514) |
| ACDC | Left ventricle | Organ | MRI | 100 (1808) | 50 (977) |
|  | Myocardium |  |  | 100 (1828) | 50 (989) |
|  | Right ventricle |  |  | 100 (1558) | 50 (881) |
| CAMUS | Left ventricle | Organ | Ultrasound | 450 (8393) | 50 (875) |
|  | Myocardium |  |  |  |  |
|  | Left atrium |  |  |  |  |
| Kiser et al. (2020) | Left lung | Organ | CT | 300 (23858) | 98 (7931) |
|  | Right lung |  |  | 300 (24026) | 98 (7962) |
| Simpson et al. (2015) | Spleen | Organ | CT | 30 (832) | 11 (219) |
| Pancreas-CT | Pancreas | Organ | CT | 60 (5158) | 20 (1724) |
| BTCV | Aorta | Organ | CT | 18 (1215) | 12 (827) |
|  | Left kidney |  |  | 18 (543) | 12 (362) |
|  | Right kidney |  |  | 18 (547) | 12 (350) |
|  | Liver |  |  | 18 (911) | 12 (631) |
|  | Spleen |  |  | 18 (532) | 12 (332) |
|  | Stomach |  |  | 18 (594) | 12 (421) |
|  | Pancreas |  |  | 18 (430) | 12 (316) |
|  | Gallbladder |  |  | 17 (228) | 11 (131) |
| Jiang et al. (2024) | Prostate | Organ | Ultrasound | 55 (1931) | 20 (690) |
| BraTS 2019 | Brain tumor | Lesion | MRI | 250 (16535) | 85 (5613) |
| Zhang et al. (2023) | Breast mass | Lesion | MRI | 80 (2913) | 20 (565) |
| RIDER |  |  |  | 3 (39) | 1 (32) |
| LiTS | Liver tumor | Lesion | CT | 20 (703) | 12 (1110) |
| KiTS 2023 | Kidney tumor | Lesion | CT | 285 (4659) | 40 (1110) |
| CVC-ClinicDB | Polyp | Lesion | Endoscopy | 18 (367) | 11 (245) |
| CVC-ColonDB |  |  |  | 6 (180) | 6 (120) |
| ETIS |  |  |  | 20 (152) | 6 (44) |
| ASU-Mayo |  |  |  | 8 (2701) | 2 (1155) |

2018), BraTS 2019 (Menze et al., 2014; Bakas et al., 2017; Baid et al., 2021), Breast Cancer DCE-MRI Data (Zhang et al., 2023), and RIDER (Meyer et al., 2015). 2) CT datasets. Thoracic cavity segmentation dataset introduced by (Aerts et al., 2019), spleen segmentation dataset introduced by (Simpson et al., 2015), Pancreas-CT (Roth et al., 2015), the abdomen part of BTCV (Landman et al., 2015), LiTS (Bilic et al., 2023), and KiTS 2023 (Heller et al., 2021; 2023), 3) Ultrasound datasets. CAMUS (Leclerc et al., 2019), which is also known as echocardiography, and Micro-Ultrasound Prostate Segmentation Dataset (Jiang et al., 2024). 4) Endoscopy datasets. CVC-ClinicDB (Bernal et al., 2015), CVC-ColonDB (Bernal et al., 2012), ETIS (Silva et al., 2014), and ASU-Mayo (Tajbakhsh et al., 2015). We maintain the original training and testing splits, and ensure that each sequence is only used in one split. Segmentation performance is evaluated using the Dice coefficient and Hausdorff distance as metrics.

## 4.2 IMPLEMENTATION DETAILS

For all datasets, we convert videos into frames and 3D volumes into 2D slices. Images without a valid object are filtered out. In total, 3,644 sequences are used in training and 1,061 sequences for testing. Data augmentation techniques include random horizontal flip, random resize, random crop, and photometric distortion. All images are resized to a maximum length of 640 pixels. The coefficients for the losses are set as $\lambda_{L1} = 5$, $\lambda_{giou} = 2$, $\lambda_{dice} = 5$, $\lambda_{focal} = 2$, and $\lambda_{cls} = 2$. We adopt 4 encoder layers and 4 decoder layers in the Transformer, and the initial query number $N_q$ is set to 5. Both the hidden dimension of the Transformer and the channel dimension of text prompts are $C = 256$. During training, 3 temporal images from a sequence are randomly sampled and fed into the model at each iteration. Our model is trained on 2 RTX 3090 24GB GPUs with a batch size of 1 per GPU, using AdamW optimizer with an initial learning rate of $10^{-5}$ for 5 epochs. The learning rate decays by 0.1 at the $3^{rd}$ epoch.

Table 2: Comparison results with state-of-the-art methods on organs. ↑ denotes higher is better and ↓ denotes lower is better. Numbers in **bold** represent the best and underlined ones are the second best. [1]Average of ACDC and CAMUS, [2]Average of left lung and right lung, [3]Average of BTCV, Pancreas-CT, and Spleen segmentation dataset (Simpson et al., 2015).

| Method | Backbone | Heart[1] | | Lung[2] | | Abdomen[3] | | Prostate | | Overall | |
|---|---|---|---|---|---|---|---|---|---|---|---|
| | | Dice↑ | HD↓ | Dice↑ | HD↓ | Dice↑ | HD↓ | Dice↑ | HD↓ | Dice↑ | HD↓ |
| URVOS | ResNet-50 | 83.92 | 3.87 | 84.61 | 5.64 | 60.19 | 4.64 | 91.92 | **10.48** | 73.07 | 4.72 |
| ReferFormer | ResNet-50 | 86.29 | 3.92 | 84.19 | 5.03 | 72.12 | 4.21 | 89.79 | 11.30 | 79.51 | 4.51 |
| OnlineRefer | ResNet-50 | 83.93 | 3.94 | 85.27 | 4.89 | 63.48 | 4.59 | 91.69 | 10.98 | 74.69 | 4.68 |
| Ours | ResNet-50 | **87.19** | **3.79** | **88.77** | **4.04** | **72.80** | **4.07** | **93.13** | 10.75 | **80.77** | **4.28** |
| ReferFormer | Swin-L | 84.12 | 3.99 | 82.56 | 5.12 | 66.05 | **4.31** | 90.58 | 11.26 | 75.67 | 4.60 |
| OnlineRefer | Swin-L | 84.37 | 3.90 | 83.59 | 4.99 | 60.39 | 4.62 | 90.72 | **10.80** | 73.30 | 4.68 |
| Ours | Swin-L | **84.47** | **3.87** | **84.96** | 4.99 | **66.41** | 4.52 | **91.54** | 10.93 | **76.25** | 4.62 |
| SOC | V-Swin-T | 81.76 | 4.22 | 84.84 | 4.94 | 62.55 | 4.82 | 86.42 | 12.73 | 73.12 | 4.98 |
| MTTR | V-Swin-T | 84.80 | 3.98 | 84.92 | 4.94 | 64.23 | 4.39 | 89.96 | 11.95 | 75.26 | 4.65 |
| Ours | V-Swin-T | **84.98** | **3.85** | **85.19** | **4.93** | **65.57** | **4.31** | **92.34** | **10.70** | **76.11** | **4.50** |

Table 3: Comparison results with state-of-the-art methods on lesions. [1]Average of Breast Cancer DCE-MRI Data and RIDER. [2]Average of CVC-ClinicDB, CVC-ColonDB, ETIS, and ASU-Mayo.

| Method | Backbone | Brain tumor | | Breast mass[1] | | Liver tumor | | Kidney tumor | | Polyp[2] | |
|---|---|---|---|---|---|---|---|---|---|---|---|
| | | Dice↑ | HD↓ | Dice↑ | HD↓ | Dice↑ | HD↓ | Dice↑ | HD↓ | Dice↑ | HD↓ |
| URVOS | ResNet-50 | 74.59 | 4.73 | 55.91 | 5.20 | 27.43 | 8.51 | 72.24 | 5.63 | 66.17 | 7.79 |
| ReferFormer | ResNet-50 | 76.60 | 3.16 | 60.70 | 4.93 | 47.43 | 8.97 | 61.75 | 6.83 | 62.75 | 8.19 |
| OnlineRefer | ResNet-50 | 77.55 | 3.00 | 64.81 | **4.48** | 39.70 | 8.85 | 74.75 | 5.58 | 72.77 | 7.31 |
| Ours | ResNet-50 | **78.24** | **2.96** | 65.40 | 4.66 | **65.27** | 6.82 | **77.73** | 5.56 | **75.56** | **7.07** |
| ReferFormer | Swin-L | 76.89 | 3.06 | 61.53 | 4.78 | 57.43 | 7.48 | 78.31 | 5.46 | 67.35 | 7.81 |
| OnlineRefer | Swin-L | 77.46 | **2.97** | 57.22 | 4.62 | 54.50 | 7.57 | 69.91 | 6.04 | **78.47** | **6.80** |
| Ours | Swin-L | **77.96** | 3.03 | **65.90** | 4.49 | **59.32** | 7.45 | **79.27** | 5.26 | 77.56 | 7.25 |
| SOC | V-Swin-T | 75.55 | 3.05 | **61.57** | 4.75 | 35.30 | 8.42 | 70.01 | 6.08 | 60.04 | 8.73 |
| MTTR | V-Swin-T | 76.21 | 3.00 | 57.74 | 4.95 | 53.68 | **7.28** | 67.31 | 6.33 | 71.12 | 7.72 |
| Ours | V-Swin-T | **77.37** | **2.98** | 59.17 | **4.52** | 54.26 | 8.55 | **76.07** | 5.61 | 77.11 | 6.93 |

## 4.3 RESULTS

**Comparison to the State-of-the-art.** We compare our method with state-of-the-art approaches on referring video object segmentation, including URVOS (Seo et al., 2020), ReferFormer (Wu et al., 2022), OnlineRefer (Wu et al., 2023), MTTR (Botach et al., 2022), and SOC (Luo et al., 2024). The comparison results for organs and lesions are shown in Table 2 and Table 3, respectively. To better organize and present the datasets, we categorize the organ datasets into four distinct groups: heart, lung, abdomen, and prostate. We then compute the average metrics for each group, allowing us to identify strengths and weaknesses specific to different anatomical regions. Detailed experimental results for each category can be found in Appendix A.2.

For feature extraction, we implement various visual backbones, including ResNet (He et al., 2016), Swin Transformer (Liu et al., 2021), and Video Swin Transformer (Liu et al., 2022). Notably, the performance for organ detection is superior to that for lesion detection. This discrepancy can be attributed to the smaller size and more homogeneous appearance of lesions, which makes them inherently more challenging to identify. Our approach consistently outperforms previous methods across all three backbones, especially on lesion datasets. For the segmentation of liver tumors and kidney tumors, our model with a ResNet-50 backbone achieves average Dice scores of 65.27% and 77.73%, which are 17.84 and 15.98 points higher than the previous state-of-the-art work, Refer-Former. Figure 5 shows the visualization results of our TPP.

**Comparison to SAM 2.** The Segment Anything Model 2 (Ravi et al., 2024) serves as a foundational model for promptable visual segmentation in images and videos. As it currently lacks support for text prompts, we utilize a community-developed version, Grounded SAM 2 (Liu et al., 2023), which enables video object tracking with text inputs. This model uses box outputs from Grounding DINO

Table 4: Comparison with SAM 2 series.

| Prompter + Segmenter | Metric | Organ | Lesion |
|---|---|---|---|
| Grounding DINO + SAM 2 | Dice↑ | 12.46 | 10.10 |
| | HD↓ | 17.08 | 21.05 |
| TPP + SAM 2 | Dice↑ | 53.45 | 54.55 |
| | HD↓ | 6.03 | 6.79 |
| TPP + TPP | Dice↑ | 80.77 | 72.69 |
| | HD↓ | 4.28 | 5.88 |

Table 5: Few-shot performance.

| Method | Metric | Right ventricle | Breast mass | Polyp |
|---|---|---|---|---|
| Full data | Dice↑ | 81.97 | 61.96 | 82.19 |
| | HD↓ | 3.45 | 4.57 | 6.65 |
| One-shot | Dice↑ | 75.63 | 59.88 | 81.55 |
| | HD↓ | 3.93 | 4.56 | 6.65 |
| Zero-shot | Dice↑ | 71.13 | 57.18 | 80.97 |
| | HD↓ | 4.29 | 4.60 | 6.70 |

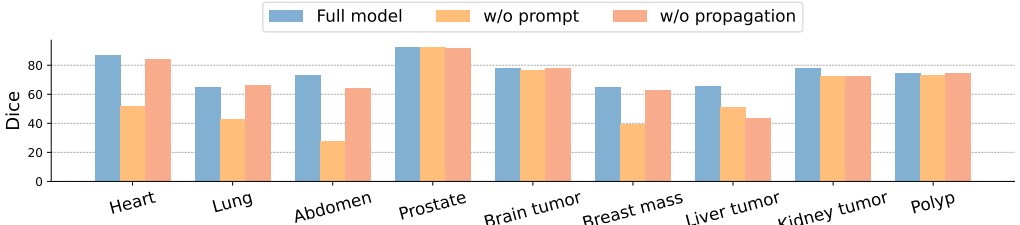

Figure 4: Ablation studies on text prompts and propagation strategies. Dice scores are provided for full model, without prompt, and without propagation, respectively.

as prompts for SAM 2's video predictor. Despite this integration, it achieves average Dice scores of only 12.46% for organs and 10.10% for lesions, indicating its limited understanding of medical text prompts. To address this, we utilize the mask predictions of the first image in the sequences generated by our TPP as mask prompts for SAM 2, which leads to substantial improvements, with average Dice scores of 53.45% (*+40.99*) for organs and 54.55% (*+44.45*) for lesions. As shown in Table 4, our TPP demonstrates superiority over Grounding DINO in text grounding ability, and surpasses SAM 2 in object tracking capabilities due to the triple propagation strategy (See Appendix A.3 for visualization results).

**Zero-/Few-shot Performance.** To validate the zero-shot performance of our approach on unseen datasets, we exclude RVSC (right ventricle), RIDER (breast mass), and CVC-ColonDB (polyp) from the training datasets and evaluate the trained model on these datasets directly. As shown in Table 5, the Dice scores for breast mass and polyp decrease by only 4.78 and 1.22 points compared to full-data training. In the one-shot setting, we use a single sequence from each of the three datasets mentioned above for training. The results show that one-shot performance is comparable to full-data training, highlighting the model's robust generalization ability.

## 4.4 ABLATION STUDY

Cross-modal prompt fusion and the propagation strategy are critical components of our approach to referring medical image sequence segmentation. Figure 4 illustrates that medical text prompts are particularly essential for accurately identifying organs located in the heart, lungs, and abdomen. Moreover, for extremely small lesions, such as breast masses and liver tumors, our propagation strategy significantly reduces the occurrence of false negatives, resulting in significant enhancements.

Table 6: Ablation studies on prompts.

| Prompt | Organ | | Lesion | |
|---|---|---|---|---|
| | Dice↑ | HD↓ | Dice↑ | HD↓ |
| w/o prompt | 41.45 | 8.14 | 63.69 | 6.55 |
| w/ [profile] | 76.17 | 4.71 | 66.07 | 6.18 |
| w/ [color] &[shape] | 78.31 | 4.70 | 67.50 | 6.34 |
| Full model | **80.77** | **4.28** | **72.69** | **5.88** |

Table 7: Ablation studies on propagation.

| Propagation | Organ | | Lesion | |
|---|---|---|---|---|
| | Dice↑ | HD↓ | Dice↑ | HD↓ |
| w/o prop. | 74.53 | 4.75 | 63.97 | 6.51 |
| w/o query | 77.86 | 4.66 | 64.03 | 6.42 |
| w/o mask | 77.93 | 4.61 | 67.10 | 6.40 |
| w/o box | 79.57 | 4.48 | 71.43 | 6.05 |
| Full model | **80.77** | **4.28** | **72.69** | **5.88** |

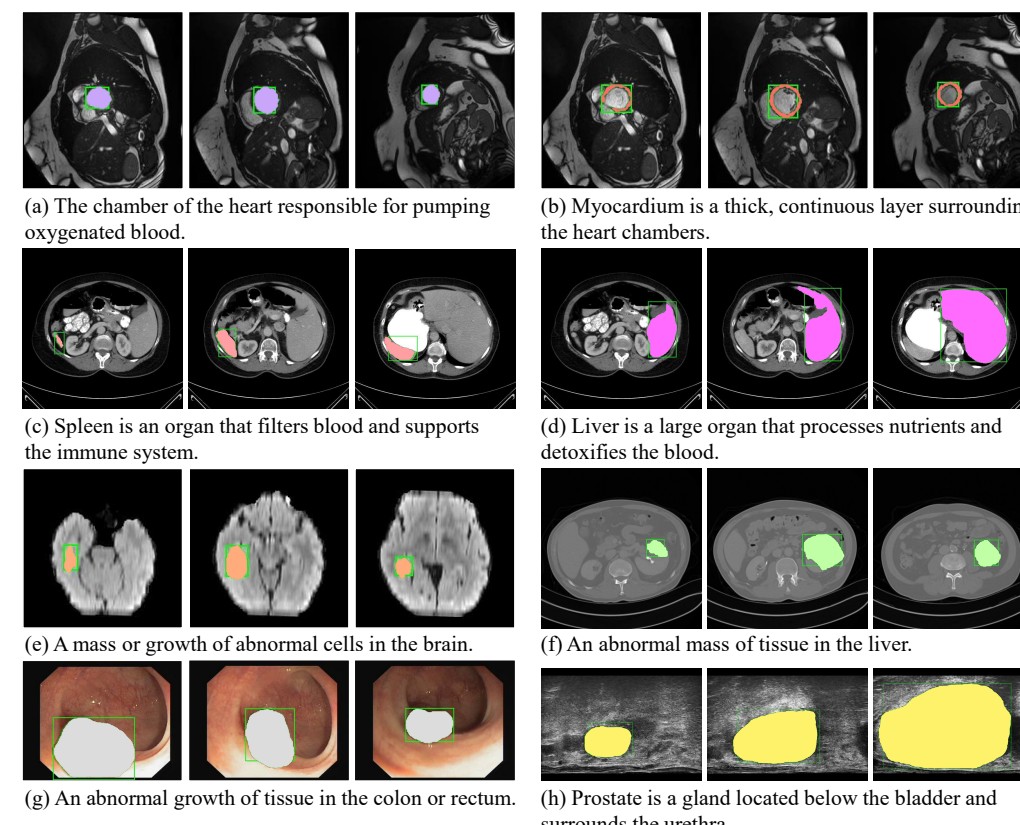

Figure 5: Visualization of segmentation results for different entities and modalities. (a) and (b) display the results of left atrium and myocardium in the same MRIs, respectively. (c) and (d) show spleen and liver in the same CT slices, respectively. From (e) to (h), visualizations are brain tumor in MRI, liver tumor in CT, polyp in endoscopy, and prostate in ultrasound.

**Medical Text Prompts.** We utilize large language models to generate three attributes for each anatomical entity: [profile], [color], and [shape]. Among these, [profile] is a more abstract concept, whereas [color] and [shape] are more specific. These different attributes serve as varied prompt messages, resulting in distinct enhancements in segmentation performance, as shown in Table 6.

**Propagation Strategy.** To investigate the effects of box propagation, mask propagation, and query propagation, we conduct ablation experiments by removing the corresponding propagation methods, as demonstrated in Table 7. The absence of mask propagation and query propagation results in decreases of 2.84 and 2.91 points in Dice score, and increases of 0.33 and 0.38 in Hausdorff distance for organs. More details on ablation studies for propagation can be found in Appendix A.4.

## 5 CONCLUSION

In this paper, we introduce a new task, termed Referring Medical Image Sequence Segmentation, accompanied by a large and comprehensive benchmark. The benchmark includes 20 different anatomical entities across 4 modalities from various regions of the body. We present an innovative text-promptable approach that effectively leverages the inherent sequential relationships and textual cues within medical image sequences to segment referred objects, serving as a strong baseline for this task. By integrating both 2D and 3D medical images through a triple-propagation strategy, we demonstrate significant improvements across a broad spectrum of medical datasets, emphasizing the potential for rapid response in segmenting referred objects and enabling accurate diagnosis in clinical practice. Future work should delve deeper into optimizing prompts and exploring additional modalities to further enhance the efficacy of medical image analysis.

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

## A APPENDIX

### A.1 MEDICAL TEXT PROMPTS

We employ GPT-4 to automatically generate medical text prompts for each anatomical entity.

---

**Instruction**

```
You are a medical expert.  Describe the [profile], [color], and [shape] of the
left atrium / right ventricle / myocardium / ...  on { MRI / CT / Ultrasound /
Endoscopy } in one sentence each.
```

**Output**

```
 Left atrium:
"the left atrium is a chamber of the heart."
"it typically appears as a lighter region."
"it is always surrounded by its walls."

 Right ventricle:
"the chamber of the heart responsible for pumping deoxygenated blood."
"the right ventricle typically appears as a brighter or lighter region adjacent to
the right atrium."
"it appears as a distinct, usually crescent-shaped structure."

 Myocardium:
"the myocardium is the muscular tissue of the heart."
"it typically appears as a darker or medium gray region."
"it is a thick, continuous layer surrounding the heart chambers."

 Left ventricle:
"the chamber of the heart responsible for pumping oxygenated blood."
"the left ventricle typically appears as a darker or medium gray region adjacent
to the left atrium."
"the left ventricle appears as a round or long structure."

 Prostate:
"the prostate is a gland located below the bladder and surrounds the urethra."
"on ultrasound, it typically appears as a gray or medium gray structure."
"the prostate usually has a somewhat rounded or oval shape."

 Brain tumor:
"a mass or growth of abnormal cells in the brain."
"brain tumors can be benign or malignant."
"exp":  "brain tumors can be round, oval, or irregular in shape."

 Liver tumor:
"a liver tumor is an abnormal mass of tissue in the liver, potentially cancerous
or benign."
"on ct, liver tumors typically appear as areas of varying density, often with
contrasting colors compared to the surrounding liver tissue."
"liver tumors can have irregular, lobulated, or well-defined shapes."

 Kidney tumor:
"a kidney tumor is an abnormal growth of cells within the kidney."
"kidney tumors often appear as areas of varying density, typically with
contrasting colors compared to the surrounding kidney tissue."
"kidney tumors can have irregular, nodular, or well-defined shapes."

 Polyp:
"an abnormal growth of tissue in the colon or rectum."
```

"the color of a colorectal polyp are often pink, red, or tan."
"colorectal polyps can have flat, raised, or mushroom-like structures."

Breast mass:
"a breast mass is an abnormal growth or lump in the breast tissue."
"breast masses typically appear as areas of variable signal intensity."
"breast masses can have irregular, lobulated, or well-defined shapes."

Aorta:
"the aorta is a large artery that carries blood from the heart to the rest of the body."
"on ct, the aorta appears as a tubular structure with varying density, often highlighted with contrast."
"the aorta typically has a cylindrical or slightly curved shape."

Gallbladder:
"the gallbladder is a small organ that stores bile."
"the gallbladder usually appears as a dark, fluid-filled sac."
"the gallbladder typically has a pear-shaped or oval shape."

Left kidney:
"the left kidney is an organ that filters blood and produces urine."
"on ct, the left kidney appears as a bean-shaped structure with variable density."
"the left kidney typically has a slightly curved or oval shape."

Right kidney:
"the right kidney is an organ that filters blood and produces urine."
"on ct, the right kidney appears as a bean-shaped structure with variable density."
"the right kidney typically has a slightly curved or oval shape."

Liver:
"the liver is a large organ that processes nutrients and detoxifies the blood."
"on ct, the liver appears as a dense, homogeneous structure with contrast-enhanced areas."
"the liver typically has a roughly triangular or irregular shape."

Pancreas:
"the pancreas is an organ that produces digestive enzymes and hormones."
"on ct, the pancreas appears as a soft tissue structure with variable density, often with contrast enhancement."
"the pancreas typically has an elongated, somewhat irregular shape."

Spleen:
"the spleen is an organ that filters blood and supports the immune system."
"on ct, the spleen appears as a soft tissue structure with homogeneous density."
"the spleen typically has an oval or crescent-shaped structure."

Stomach:
"the stomach is an organ that digests food and stores it before it moves to the intestines."
"on ct, the stomach appears as a variable density structure with contrast-enhanced areas."
"the stomach typically has a j-shaped or irregular shape."

Left lung:
"the left lung is an organ that facilitates breathing and gas exchange."
"the color of the left lung typically appears as a grey or slightly darker compared to the surrounding tissues."
"the shape of the left lung is generally asymmetrical, often appearing somewhat triangular or wedge-shaped."

Right lung:
"the right lung is an organ that facilitates breathing and gas exchange."
"the color of the right lung typically appears as a grey or slightly darker compared to the surrounding tissues."
"the shape of the right lung is generally asymmetrical, often appearing somewhat triangular or wedge-shaped."

Table 8: Detailed comparison results with state-of-the-art methods. ↑ denotes higher is better, and ↓ denotes lower is better. Numbers in **bold** represent the best and underlined ones are the second best. [1]ACDC, [2]CAMUS, [3]BTCV, [4]RIDER, [5]CVC-ClinicDB, [6]CVC-ColonDB, [7]ETIS, [8]ASU-Mayo.

| Class | Method Backbone | URVOS R-50 | ReferFormer R-50 | ReferFormer Swin-L | OnlineRefer R-50 | OnlineRefer Swin-L | MTTR V-Swin-T | SOC V-Swin-T | Ours R-50 | Ours Swin-L | Ours V-Swin-T |
|---|---|---|---|---|---|---|---|---|---|---|---|
| Left atrium | Dice↑ | 77.84 | 82.27 | 80.87 | 80.02 | 80.12 | 81.81 | 81.40 | **83.05** | 74.68 | 79.17 |
| | HD↓ | 3.00 | 3.76 | 3.92 | 3.96 | 3.99 | 3.90 | 3.99 | 3.72 | 4.02 | 3.94 |
| Right ventricle | Dice↑ | 76.34 | 77.93 | 81.39 | 73.88 | 80.83 | 77.21 | 76.25 | 81.97 | **83.63** | 79.72 |
| | HD↓ | 4.51 | 3.84 | 3.39 | 3.76 | 3.37 | 3.73 | 4.06 | 3.45 | 3.29 | 3.44 |
| Left ventricle[1] | Dice↑ | 87.12 | 90.57 | 88.36 | 87.26 | 85.92 | 89.58 | 89.57 | **90.14** | 86.45 | 87.54 |
| | HD↓ | 3.48 | 2.02 | 2.03 | 2.13 | 2.13 | 2.04 | 2.08 | 2.04 | 2.10 | 2.08 |
| Myo-cardium[1] | Dice↑ | 78.55 | 82.92 | 76.49 | 80.98 | 79.98 | 78.62 | 57.27 | **84.34** | 78.19 | 79.76 |
| | HD↓ | 3.53 | 2.74 | 2.88 | 2.84 | 2.82 | 2.82 | 4.29 | **2.68** | 2.89 | 2.81 |
| Right ventricle[1] | Dice↑ | 81.06 | 85.47 | 82.08 | 79.08 | 79.68 | 83.97 | 81.63 | **85.77** | 83.77 | 83.22 |
| | HD↓ | 3.55 | 2.72 | 2.76 | 2.94 | 2.90 | 2.73 | 2.94 | **2.68** | 2.80 | 2.75 |
| Left ventricle[2] | Dice↑ | 93.23 | 92.69 | 91.98 | 93.11 | 92.70 | 92.97 | 92.40 | **93.50** | 92.24 | 92.93 |
| | HD↓ | 4.19 | 4.74 | 5.04 | 4.72 | 4.77 | 4.63 | 4.72 | 4.63 | 4.76 | 4.71 |
| Myo-cardium[2] | Dice↑ | 88.57 | 88.17 | 85.53 | 88.39 | 88.17 | 87.07 | 87.05 | **89.03** | 86.39 | 88.10 |
| | HD↓ | 4.87 | 6.36 | 6.45 | 6.09 | 6.12 | 6.47 | 6.39 | 6.07 | 6.20 | 6.06 |
| Left atrium[2] | Dice↑ | 88.69 | 90.28 | 86.30 | 88.70 | 89.94 | 87.20 | 88.48 | 89.73 | **90.45** | 89.43 |
| | HD↓ | 3.81 | 5.13 | 5.44 | 5.09 | 5.13 | 5.53 | 5.26 | 5.02 | **4.92** | 5.04 |
| Left lung | Dice↑ | 85.23 | 85.14 | 83.92 | 86.10 | 84.60 | 85.02 | 85.97 | **89.88** | 87.38 | 86.65 |
| | HD↓ | 5.67 | 5.01 | 5.15 | 4.96 | 5.00 | 4.95 | 4.95 | **4.06** | 4.95 | 4.96 |
| Right lung | Dice↑ | 84.00 | 83.23 | 81.20 | 84.43 | 82.58 | 84.83 | 83.72 | **87.65** | 82.54 | 83.72 |
| | HD↓ | 5.61 | 5.04 | 5.08 | 4.82 | 4.98 | 4.93 | 4.92 | **4.01** | 5.03 | 4.91 |
| Spleen | Dice↑ | 80.33 | 84.63 | 87.06 | 87.37 | 87.17 | 84.48 | 81.66 | **90.20** | 86.54 | 84.56 |
| | HD↓ | 4.64 | 4.29 | 3.69 | 4.04 | 3.84 | 3.88 | 4.51 | **3.69** | 3.98 | 3.79 |
| Pancreas | Dice↑ | 10.77 | 23.36 | 10.71 | 26.75 | 23.29 | 27.98 | 31.41 | 26.02 | 18.58 | 22.36 |
| | HD↓ | 5.01 | 5.64 | 5.26 | 5.17 | 5.39 | 4.99 | 5.06 | 5.05 | 5.91 | 5.02 |
| Aorta | Dice↑ | 60.81 | 88.12 | 80.84 | 82.12 | 75.16 | 83.64 | 80.22 | 86.14 | 73.63 | 75.42 |
| | HD↓ | 3.41 | 2.23 | 2.45 | 2.56 | 2.73 | 2.58 | 2.62 | 2.40 | 2.64 | 2.48 |
| Left kidney | Dice↑ | 79.16 | 89.71 | 61.06 | 77.01 | 51.57 | 68.11 | 79.75 | 87.53 | 84.23 | 81.60 |
| | HD↓ | 4.02 | 3.20 | 4.42 | 3.92 | 4.68 | 3.81 | 4.08 | 3.26 | 3.44 | 3.40 |
| Right kidney | Dice↑ | 75.82 | 84.72 | 84.00 | 68.88 | 62.53 | 65.05 | 79.05 | 84.16 | 81.36 | 80.94 |
| | HD↓ | 4.42 | 3.62 | 3.78 | 4.29 | 4.49 | 4.11 | 4.24 | **3.59** | 3.85 | 3.71 |
| Liver | Dice↑ | 85.55 | 89.18 | 88.00 | 86.75 | 87.81 | 87.54 | 85.09 | **90.32** | 88.62 | 88.89 |
| | HD↓ | 5.26 | 5.23 | 5.13 | 5.34 | 5.25 | 5.17 | 5.71 | **5.06** | 5.16 | 5.11 |
| Spleen[3] | Dice↑ | 79.58 | 88.19 | 84.45 | 84.08 | 84.68 | 85.37 | 76.36 | **88.41** | 85.34 | 82.98 |
| | HD↓ | 4.48 | 3.85 | 4.00 | 4.16 | 4.07 | 4.02 | 4.48 | **3.72** | 3.93 | 3.92 |
| Stomach | Dice↑ | 63.61 | 64.52 | 61.40 | 59.97 | 54.93 | 56.41 | 46.53 | **67.35** | 63.66 | 55.83 |
| | HD↓ | 5.54 | 5.44 | 5.70 | 5.97 | 6.19 | 6.22 | 7.10 | **5.39** | 5.96 | 6.20 |
| Pancreas[3] | Dice↑ | 27.28 | 50.50 | 41.53 | 33.66 | 38.40 | 40.13 | 35.59 | 47.61 | 42.36 | 39.37 |
| | HD↓ | 4.99 | 4.51 | 4.72 | 5.12 | 4.90 | 4.88 | 5.11 | 4.69 | 5.34 | 4.89 |
| Gall-bladder | Dice↑ | 39.00 | 58.23 | 61.42 | 28.17 | 38.44 | 43.59 | 29.88 | **60.29** | 39.78 | 43.79 |
| | HD↓ | 4.60 | 4.03 | 3.98 | 5.36 | 4.65 | 4.28 | 5.26 | **3.94** | 4.95 | 4.55 |
| Prostate | Dice↑ | 91.92 | 89.79 | 90.58 | 91.69 | 90.72 | 89.96 | 86.42 | **93.13** | 91.54 | 92.34 |
| | HD↓ | 10.48 | 11.30 | 11.26 | 10.98 | 10.79 | 11.95 | 12.73 | 10.75 | 10.93 | 10.70 |
| Brain tumor | Dice↑ | 74.59 | 76.60 | 76.89 | 77.55 | 77.46 | 76.21 | 75.55 | **78.24** | 77.96 | 77.37 |
| | HD↓ | 4.72 | 3.16 | 3.06 | 3.00 | 2.97 | 3.00 | 3.05 | **2.96** | 3.03 | 2.98 |
| Breast mass | Dice↑ | 60.85 | 63.12 | 66.06 | 67.91 | 65.39 | 70.04 | 63.92 | 68.83 | 67.00 | 61.02 |
| | HD↓ | 5.15 | 5.19 | 5.16 | 4.92 | 4.90 | 4.91 | 4.82 | **4.75** | 4.75 | 5.02 |
| Breast mass[4] | Dice↑ | 50.97 | 58.28 | 57.00 | 61.71 | 49.05 | 45.44 | 59.22 | **61.96** | 64.80 | 57.32 |
| | HD↓ | 5.26 | 4.66 | 4.41 | 4.05 | 4.33 | 4.99 | 4.68 | 4.57 | 4.23 | **4.02** |
| Liver tumor | Dice↑ | 27.43 | 47.43 | 57.43 | 39.70 | 54.50 | 53.68 | 35.30 | 65.27 | 59.32 | 54.26 |
| | HD↓ | 8.51 | 8.97 | 7.48 | 8.85 | 7.57 | 7.28 | 8.42 | **6.82** | 7.45 | 8.55 |
| Kidney tumor | Dice↑ | 72.24 | 61.75 | 78.31 | 74.75 | 69.91 | 67.31 | 70.01 | 77.73 | **79.27** | 76.07 |
| | HD↓ | 5.63 | 6.83 | 5.46 | 5.58 | 6.04 | 6.33 | 6.08 | 5.56 | **5.26** | 5.61 |
| Polyp[5] | Dice↑ | 80.90 | 75.65 | 80.74 | 81.81 | 85.19 | 78.85 | 70.28 | 81.24 | **85.46** | 82.13 |
| | HD↓ | 8.01 | 5.59 | 5.12 | 5.36 | 5.03 | 6.03 | 6.73 | 5.28 | 5.12 | 5.33 |
| Polyp[6] | Dice↑ | 77.98 | 79.94 | 80.02 | 69.75 | 84.41 | 77.49 | 75.16 | 82.19 | **84.02** | 79.83 |
| | HD↓ | 8.19 | 6.53 | 6.54 | 7.39 | 6.24 | 7.11 | 7.32 | 6.66 | 6.30 | 6.69 |
| Polyp[7] | Dice↑ | 51.48 | 60.16 | 63.64 | 67.65 | 67.27 | 57.70 | 54.38 | 65.58 | **72.35** | 68.83 |
| | HD↓ | 8.29 | 10.37 | 10.07 | 9.54 | 9.36 | 10.55 | 10.97 | 9.45 | 9.36 | **9.19** |
| Polyp[8] | Dice↑ | 54.30 | 35.27 | 44.98 | 71.89 | 77.04 | 70.46 | 40.33 | 73.22 | 68.41 | 77.63 |
| | HD↓ | 6.65 | 10.29 | 9.50 | 6.95 | 6.58 | 7.19 | 9.88 | 6.88 | 8.21 | **6.50** |

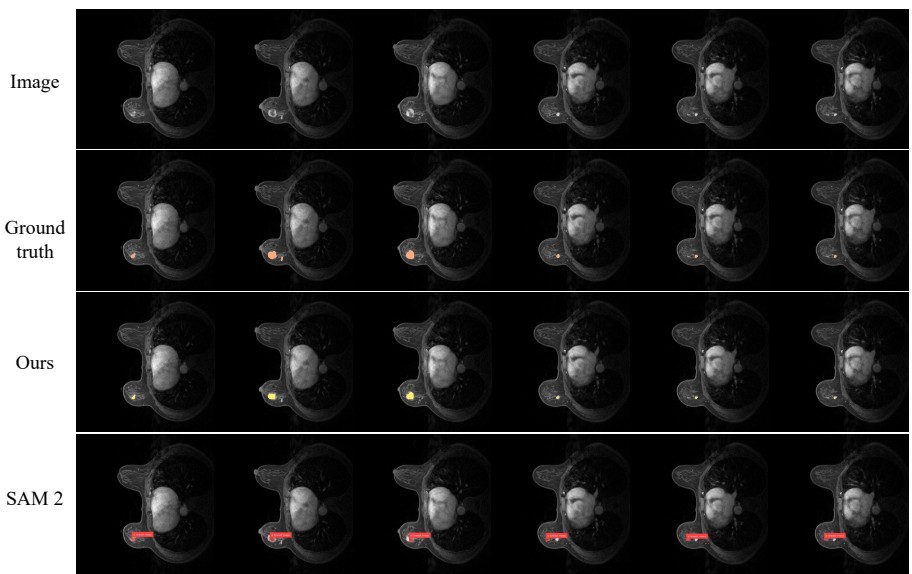

Figure 6: Qualitative comparison results with SAM 2. Zoom in for details.

## A.2    DETAILED EXPERIMENTAL RESULTS

We use two metrics for medical image sequence segmentation. Let $M$ and $Y$ be the predicted masks and ground truth masks, respectively, and let $m$ and $y$ be the corresponding contours delineating the object. The following two standard similarity measurements are computed:

• Dice: It measures the overlap or similarity between two masks and is defined as:

$$\mathcal{D}(M,Y) = 2\frac{M \cap Y}{M + Y} \tag{15}$$

• Hausdorff distance: It is a symmetric measure of distance between two contours and is defined as:

$$\mathcal{H}(m,y) = \max\left(\max_{i \in m}\left(\min_{j \in y} d(i,j)\right), \max_{j \in y}\left(\min_{i \in m} d(i,j)\right)\right) \tag{16}$$

For clarity and due to page limitations, we only report the average metrics for datasets of human body parts in the main text. Here, we present the detailed metrics for each category in Table 8. Our TPP demonstrates superior performance, particularly in the segmentation of heart structures, lungs, masses, and tumors. On the BTCV dataset, ReferFormer exhibits excellent results. Among the three backbone architectures, ResNet-50 proves to be the most robust across the majority of tasks, while Swin Transformer (Large) excels specifically in polyp segmentation.

## A.3    MORE VISUALIZATION RESULTS

Figure 6 provides a qualitative comparison between our TPP and SAM 2. Although we provide mask prompts as a strong initialization for SAM 2, it incorrectly identifies the cavity inside the tumor and loses track of the target in the later stages of the sequence due to the absence of text prompts.

As shown in Figure 7, both (a) and (b) fail to locate the entire liver tumor. In poorly performing cases, the segmentation either misses critical portions of the tumor or incorrectly identifies surrounding tissue as part of the lesion. This highlights the challenges in detecting complex or irregularly shaped tumors, especially in low-contrast CT scans.

## A.4    PROPAGATION STRATEGY ANALYSIS

Table 9 presents a detailed comparison of the performance improvements driven by box propagation, mask propagation, and query propagation. The results indicate that box propagation yields the

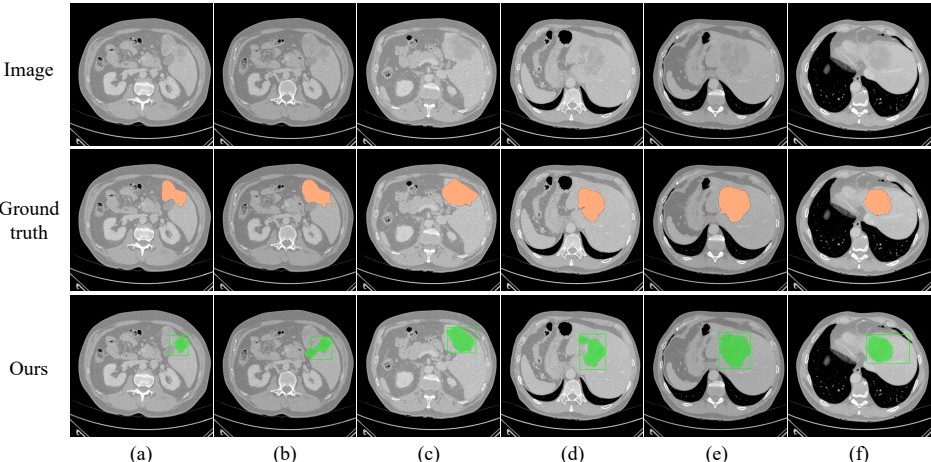

Figure 7: Poorly performing samples. Best viewed in color.

smallest enhancements, with increases of 1.16 points for organs and 2.80 points for lesions in Dice scores. In contrast, mask and query propagation demonstrate a more significant impact, highlighting their critical roles in improving overall segmentation performance. This underscores the importance of designing appropriate propagation methods to optimize results in medical image sequence segmentation.

Table 9: Ablation studies on triple propagation.

| Box propagation | Mask propagation | Query propagation | Organ Dice↑ | Organ HD↓ | Lesion Dice↑↑ | Lesion HD↓ |
|---|---|---|---|---|---|---|
| | | | 74.53 | 4.75 | 63.97 | 6.51 |
| ✓ | | | 76.69 | 4.78 | 66.77 | 6.29 |
| | ✓ | | 77.10 | 4.73 | 70.03 | 5.94 |
| | | ✓ | 77.28 | 4.44 | 69.37 | 6.25 |
| ✓ | ✓ | | 77.86 | 4.66 | 64.03 | 6.42 |
| ✓ | | ✓ | 77.93 | 4.61 | 67.10 | 6.40 |
| | ✓ | ✓ | 79.57 | 4.48 | 71.43 | 6.05 |
| ✓ | ✓ | ✓ | **80.77** | **4.28** | **72.69** | **5.88** |

Table 10: Analysis on query selection.

| The number of queries for Slice 1 | Slice 2 | Slice 3 | Organ Dice↑ | Organ HD↓ | Lesion Dice↑ | Lesion HD↓ |
|---|---|---|---|---|---|---|
| 5 | 5 | 5 | 79.47 | 4.39 | 70.98 | 6.11 |
| 5 | 3 | 1 | 78.47 | 4.44 | 71.67 | 6.01 |
| 5 | 1 | 1 | **80.77** | **4.28** | **72.69** | **5.88** |

Table 10 analyses the query selection strategy. The first row represents the absence of a selection process. In the second row, the model selects the top-3 queries for Slice 2, and consequently selects the top-1 query for Slice 3. However, neither strategy outperforms the baseline.

