# OpenReview forum: "Text-promptable Propagation for Referring Medical Image Sequence Segmentation"
_ICLR.cc/2025/Conference — Submitted to ICLR 2025_

### Official Review · Reviewer_iLhU · 2024-10-26

**Soundness:** 2
**Presentation:** 3
**Contribution:** 2
**Rating:** 5
**Confidence:** 5

**Summary:**

This paper introduces medical image sequence segmentation, alongside Text-Promptable Propagation (TPP), which segments anatomical structures in sequential medical images guided by text prompts. TPP integrates cross-modal prompt fusion and a transformer-based triple propagation strategy. The method is developed for 2D and 3D medical image sequences with text-based references. For the dataset, the author curate a testbed with different imaging modalities and anatomical entities. Experimental results show that the proposed TPP is promising.

**Strengths:**

Originality
The paper introduces a novel approach in both application and methodology. The originality lies in addressing Referring Medical Image Sequence Segmentation, which combines medical imaging sequences with medical text prompts to perform segmentation. The proposed TPP)segmentation model leverages text-based guidance for segmentation across both 2D and 3D medical image sequences, creating a cross-modal approach that integrates visual and linguistic information.

Quality
The quality of the work is reinforced by its comprehensive, well-curated dataset that spans 18 diverse medical datasets across 4 modalities (MRI, CT, ultrasound, and endoscopy). The scope includes 20 different organs and lesions, encompassing a wide range of anatomical and pathological structures, which bolsters the reliability and generalizability of the proposed model.

Clarity
The paper is well-written, with a logical structure that guides the reader through the problem setup, methodology, and results. The clarity is further enhanced by comprehensive illustrations of the model architecture and segmentation results, allowing readers to follow along without extensive prior knowledge.

Significance
It explores the task of medical image sequence segmentation with text-guided prompts, addressing an essential need for context-aware segmentation in clinical settings where target structures may vary widely.
The development of the TPP segmentation model demonstrates a robust, adaptable framework that can interpret and propagate segmentation instructions across sequential medical images.
It curates a large-scale, diverse dataset specifically designed for this new task, contributing a valuable resource that could drive further research and improvements in text-promptable segmentation models for healthcare applications.

**Weaknesses:**

The results lack fairness due to the absence of comparisons with stronger SOTA baselines, such as but not limited to UNeXt, 3D UX-Net, SwinUnet, and UNetR. Current baselines in the paper are comparatively weak, which does not fully substantiate the advantage of the proposed TPP model. Specifically, for the BTCV dataset, SOTA methods have reported Dice scores close to 0.8, whereas the proposed method significantly underperforms. Adding a comparison with these robust SOTA models would provide a clearer picture of the TPP model’s effectiveness.

The proposed method may be overly complex for its purpose. As mentioned, this intricate design might not necessarily outperform simpler architectures that operate in fully or semi-supervised settings without relying on text prompts.

The paper does not clearly explain the advantages of treating 3D volumes as sequential data rather than applying a direct 3D model. A direct 3D model would likely capture context and spatial relationships more explicitly, whereas the sequential approach might weaken the model’s ability to leverage 3D spatial coherence. Not it lacks of theoretical or empirical comparison to justify the choice of sequential processing.

The value added by text prompts is unclear. The current implementation uses pseudo-text sequences, which may not provide personalized or contextually enriched guidance for segmentation. This detracts from the potential significance of using text prompts. On the other hand, if the text is from the actual clinical reports and from the same person with the image, it would make the model understand the personalized differences and add clinical value.

It is unclear whether the authors trained separate models for each of the 18 datasets or a single model with all datasets.

**Questions:**

Why not stronger baselines?

Is this method overcomplicated for medical image segmentation?

Why it is better than 3D models, even without text?

Why the general description of the organ would provide significant information for segmentation, without seeing personal radiological report?

---

> ### Author Response · Authors · 2024-11-24
> **Response to Reviewer iLhU**
>
> Thank you for your encouraging recognition of our work and constructive feedback. We have carefully addressed each of your points in detail and hope that these clarifications effectively resolve your concerns.
>
> ## [iLhU-W5] Separate models for each of the 18 datasets or a single model with all datasets?
>
> We train **a single model across all datasets** to achieve a universal solution for medical image sequence segmentation, in contrast to specialized methods that are trained for each dataset individually. Despite this challenging setup, our method achieves competitive performance.
>
> ## [iLhU-Q1, Q3, W1, W3] Comparison with stronger baselines and 3D models.
>
> Thank you very much for pointing out the area for improvement. We compared our method with  other universal methods, such as CLIP-based segmentation [1], getting competitive results (73.70% vs. 76.48%).
>
> Swin-UNet [2],  a strong 3D model which is a UNet-like Transformer for medical image segmentation, was also evaluated.  Using its official open-source code, we trained it on the BTCV dataset, achieving an average Dice score of 73.38%. Our model outperforms Swin-UNet by 3.1%, with an average Dice score of 76.48%.
>
> Our Triple propagation leverages consistency in appearance and spatial relationships across frames or slices in the temporal order of medical image sequences. This approach exhibits **strong tracking ability** while maintaining a **lower computational cost** (130.77 GFLOPs vs. 142.78 GFLOPs) compared to 3D models.
>
> - Table 1: Comparison results with stronger baselines and 3D models on abdominal organs.
>
>     | Method | Aorta | Kidney (L) | Kidney (R) | Liver | Spleen | Stomach | Pancreas | Gallbladder | Average |
>     | ----- | :-----: | :-----: | :-----: | :-----: | :-----: |  :-----: |  :-----: |  :-----: |  :-----: |
>     | CLIP-driven [1] | 88.31 | 84.94 | 81.04 | 93.03 | 79.83 | 65.88 | 42.74 | 53.85 | 73.70 |
>     | Swin-UNet [2] | 77.85 | 82.34 | 75.60 | 90.07 | 86.97 | 66.89 | 52.49 | 54.84 | 73.38 |
>     | Ours | 86.14 | 87.53 | 84.16 | 90.32 | 88.41 | 67.35 | 47.61 | 60.29 | 76.48 |
>
> - Table 2: Comparison of computational cost with 3D models.
>
>     | Method | FLOPs (G) $\downarrow$ |
>     | ----- | ----- |
>     | Swin-UNet [2] | 142.78 |
>     | Ours | 130.77 |
>
> ## [iLhU-Q2, W2, W4] Value of adding medical text prompts.
>
> The value of adding text prompts lies in clinical scenarios.
>
> - When clinicians need to diagnose  diseases that are challenging to detect, such as pneumothorax, they often **need the help of radiologists to pinpoint the location**. A text-promptable model can automate this process, streamlining workflows and improving efficiency.
> - Another critical scenario is **minimizing the risk of missed lesions**. For  example, missing  polyps during endoscopy can have severe consequences, potentially  endangering the patient's life. A text-promptable model can serve as a reminder, assisting clinicians in identifying such lesions more effectively.
>
> ## [iLhU-Q4, W4] Personalization of descriptions.
>
> We greatly value your suggestion and agree that personalized and context-specific text prompts can enhance the model's clinical applicability. As you mentioned, using text derived  from actual clinical reports and tied to the same patient as the image would enable the model to understand personalized differences, thereby adding clinical value. We are committed to **annotating detailed, patient-specific prompts for each sequence**, even when they belong to the same category. For example, "The polyp in this image is tan located at the left." or "The polyp is pink and flat." We believe such efforts will better support  the task of Referring Medical Image Sequence Segmentation.
>
> >[1] Liu, Jie, et al. "Clip-driven universal model for organ segmentation and tumor detection." ICCV 2023.
>
> >[2] Cao, Hu, et al. "Swin-unet: Unet-like pure transformer for medical image segmentation." ECCV 2022.

---

> > ### Comment · Reviewer_iLhU · 2024-11-24
> >
> > Thank you for your response to the reviewers. I have slightly lowered my rating as my concerns remain unaddressed:
> > 1. The new results are still significantly lower than state-of-the-art methods, leaving the advantages of the proposed method, especially compared to 3D models, unclear.
> > 2. The explanation of why the pseudo-text information is helpful is not entirely convincing.

---

> ### Author Response · Authors · 2024-11-25
> **Official Comment by Authors**
>
> Thank you very much for your additional feedback. Below, we address your remaining concerns regarding our performance compared to state-of-the-art (SOTA) methods and the value of pseudo-text prompts.
>
> ## 1. Performance Compared to state-of-the-art and 3D Models
> We respectfully note that our method **achieves state-of-the-art performance** when compared to strong baselines, including widely recognized 3D models such as Swin-UNet. On the BTCV dataset, our method achieves an average Dice score of 76.48%, which **outperforms Swin-UNet (73.38%) by 3.1% points**. Furthermore, our approach is computationally efficient, requiring fewer FLOPs (130.77 GFLOPs for our method vs. 142.78 GFLOPs for Swin-UNet).
>
> In addition, when compared to other universal segmentation approaches such as CLIP-driven segmentation, our method consistently **achieves superior results** (73.70% for CLIP-driven vs. 76.48% for our method on average Dice score). These improvements highlight the efficacy of our proposed framework, particularly the advantages of the triple propagation mechanism in leveraging sequence consistency.
> | Method | Dice score (average) $\uparrow$ |
> | ----- | :-----: |
> | CLIP-driven | 73.70 |
> | Swin-UNet | 73.38 |
> | Ours | **76.48** |
>
> Our method also demonstrates robustness across various datasets (15 datasets for organs and 5 datasets for lesions), confirming its **generalization ability**. While 3D models may rely on volumetric context, our results show that the combination of temporal modeling with text prompts provides stronger performance, while reducing computational overhead, making it more suitable for large-scale clinical applications.
>
> ## 2. Impact of and Pseudo-Text Prompts
> We appreciate your concerns regarding  our explanation of  text prompts. While pseudo-text lacks personalization, it offers distinct advantages:
>
> - **Performance Enhancement:** In our experiments, pseudo-text consistently improved segmentation performance, even for  challenging tasks. For example, when segmenting lesions like polyps, **the inclusion of text prompts improved the average Dice score by 9.0%** across five lesion types, including brain tumor, liver tumor, kidney tumor, breast mass, and polyp.
>
> - **Flexibility and Adaptability:** Text prompts allow the model to identify target organ or lesion under their guidance, minimizing **the risk of missed lesions**. This capability **supports radiologists and endoscopists** in  identifying nodules, polyps, and other critical abnormalities. The flexibility to **focus on specific anatomical entities**, such as the pancreas in multi-organ CT scans for pancreatic cancer diagnosis, makes our approach more practical for real-world deployment.
>
> - **Potential for Personalization:** As noted previously, we are actively exploring the incorporation of patient-specific clinical reports to further enhance the utility and clinical relevance of the model.
>
> We hope these clarifications address your remaining concerns. Your valuable suggestions are greatly appreciated.

---

> > ### Comment · Reviewer_iLhU · 2024-11-25
> >
> > Thanks for the author's feedback. Please check their performance such as, but not limited to,
> > SwinU-Net
> > https://arxiv.org/pdf/2105.05537
> > UNETR
> > https://arxiv.org/pdf/2103.10504

---

> > > ### Author Response · Authors · 2024-12-02
> > > **Official Comment by Authors**
> > >
> > > Thank you very much for your valuable feedback.
> > >
> > > The comparison results presented above are based on the following settings:
> > > 1. We re-trained Swin-UNet using its official implementation on the BTCV dataset and obtained the corresponding results.
> > > 2. Our method's **universal model design**, where a single model is trained across all 18 datasets and 20 anatomical structures. Under these universal settings, our approach demonstrates superior performance compared to both CLIP-driven [1] and Swin-UNet [2].
> > >
> > > For a fairer comparison, we followed the evaluation metrics **reported in the original Swin-UNet paper** and **re-trained our model specifically** on the BTCV dataset for the 8 abdominal organs. The results show that our method outperforms Swin-UNet in this specific task.
> > > | Method | Aorta | Left Kidney | Right Kidney | Liver | Spleen | Stomach | Pancreas | Gallbladder | Average |
> > > | ----- | :-----: | :-----: | :-----: | :-----: | :-----: |  :-----: |  :-----: |  :-----: |  :-----: |
> > > | Swin-UNet | 85.47 | 83.28 | 79.61 | 94.29 | 90.66 | 76.60 | 56.58 | 66.53 | 79.13 |
> > > | Ours | 87.85 | 89.75 | 84.40 | 91.48 | 90.78 | 71.24 | 62.33 | 67.25 | **80.64** |
> > >
> > > These results reinforce the strengths of our approach **both in universal and specific settings**. Thank you for highlighting these points, which allowed us to provide additional clarity.

---

### Official Review · Reviewer_bfV2 · 2024-10-28

**Soundness:** 2
**Presentation:** 3
**Contribution:** 2
**Rating:** 3
**Confidence:** 5

**Summary:**

This paper introduces a novel task, "Referring Medical Image Sequence Segmentation," aimed at segmenting specific anatomical structures or lesions in medical image sequences based on text prompts. To tackle this task, the authors propose a robust baseline model, Text-Promptable Propagation (TPP), which leverages temporal and cross-modal relationships within image sequences to achieve precise segmentation guided by text prompts. The key contributions include a cross-modal prompt fusion technique that integrates text and image information and a Transformer-based triple-propagation strategy that utilizes spatial and temporal consistency for accurate object tracking across sequences. Additionally, the authors curated a comprehensive benchmark dataset, Ref-MISS, covering diverse imaging modalities and anatomical entities, and demonstrated the superior performance of the TPP model through extensive experiments.

**Strengths:**

1. The paper is well-structured and presents a novel approach to medical image sequence segmentation, making it overall clear and logically organized.
2. The method presented in this paper is quite innovative, particularly in its use of cross-modal prompt fusion and triple-propagation techniques for referring medical image sequence segmentation.
3. The Medical image sequence datasets covers 4 modalities and 20 anatomical entities, which is a large and relatively comprehensive.

**Weaknesses:**

1.  A notable concern is the assumption that 3D imaging slices and video frames can be processed uniformly. While this may be technically feasible, it raises questions about practical applicability since 3D slices are typically evenly spaced, whereas video frames are often sampled or held at irregular intervals. This discrepancy might impact real-world usability, especially when temporal consistency is essential. A deeper analysis or justification of this choice, addressing its implications for varied temporal resolutions, would strengthen the method’s practical relevance.
2. The method adds the use of text prompts for segmentation, but it's worth questioning the added value of prompts given the effectiveness of fully supervised segmentation methods in medical imaging. Current fully supervised models achieve high accuracy without the need for additional prompts, especially in standardized medical datasets. The practical advantage of integrating prompts is not fully addressed, and further justification is needed to clarify whether prompts enhance segmentation accuracy, adaptability, or clinical interpretability in meaningful ways beyond what traditional supervised models provide.
3. The use of three prediction heads (box, mask, and class) in the proposed model is an interesting design choice, but the technical rationale for including both box and mask heads could be further clarified. Since the mask head inherently provides pixel-level precision, it seems redundant to have a box head, as the bounding box is generally a less precise representation. A deeper explanation of the box head’s role, particularly regarding how it contributes to the model's performance or stability during training, would be valuable. For example, if the box head aids in providing a coarse location as an initial reference for mask prediction, or if it enhances the model's ability to generalize across various object sizes, this should be explained to justify its inclusion alongside the mask head.
4. The selection of comparison methods in the experiments lacks representation of the latest state-of-the-art models. Notably, there is no comparison with recent benchmark methods like nnU-Net, which is widely recognized for its performance in medical image segmentation. Including nnU-Net or other recent high-performing models as baselines would provide a more robust evaluation and better demonstrate the advantages of the proposed method over current state-of-the-art techniques. This would enhance the credibility of the performance claims and place the proposed model’s effectiveness in a more competitive context.
5. The prompt experiments are a crucial aspect of this study, as they demonstrate the effectiveness and added value of incorporating prompts. However, the current experimental setup for prompt evaluation is relatively simple. Expanding these experiments would be beneficial, perhaps by examining various prompt types, specificity levels, or prompt designs to assess their impact on segmentation accuracy and adaptability. Additionally, testing on different anatomical structures or datasets could provide insight into how prompts contribute under varied conditions. This expanded exploration would strengthen the argument for using prompts and provide a clearer understanding of their practical advantages.
6. Some areas of expression contain minor ambiguities that could benefit from clarification. For example, terminology like "the referred object" (P5, L162-166) may not be immediately clear to readers, and consistency in using terms such as "Referring Medical Image Sequence Segmentation" would improve readability.

**Questions:**

I'm a bit puzzled about the clinical significance of the new task proposed in this paper, 'Referring Medical Image Sequence Segmentation.' When performing a referring task, a physician typically already has preliminary information on the presence of certain lesions or diseases within the image sequence. This scenario seems somewhat inconsistent with the actual workflow of radiologists when conducting diagnostic assessments. Therefore, what is the medical relevance of the proposed task, 'Referring Medical Image Sequence Segmentation,' and in what specific scenarios could it be practically applied?

---

> ### Author Response · Authors · 2024-11-24
> **(1/2) Response to Reviewer bfV2**
>
> Thank you for your encouraging recognition of our work and constructive feedback. We have carefully addressed each of your points in detail and hope that these clarifications effectively resolve your concerns.
>
> ## [bfV2-Q1] What are the clinical scenarios and medical relevance of the proposed task?
>
> In clinical scenarios, text-promptable method offers distinct advantages across various user groups by addressing key needs and enhancing workflows:
> 1. **For Radiologists:** The method serves as an efficient **double-check** mechanism, especially in high-stakes scenarios where subtle lesions might be missed due to fatigue or high workload. By automating the detection and segmentation of potential areas of concern, the model assists radiologists in validating their diagnoses and avoiding missed lesions.
> 2. **For Clinicians with Limited Imaging Expertise:** Many clinicians, such as general practitioners or specialists outside radiology, lack advanced training in interpreting complex imaging studies. Text-promptable segmentation bridges this expertise gap by providing clear visual cues and annotations to help identify critical findings. For example, the system can **highlight the location of a pulmonary nodule based on a textual description**, acting as a guide and saving valuable time during decision-making. This approach builds a stronger connection between radiologists and clinicians, fostering collaborative and effective patient care.
> 3. **For Patients:** Simple and intuitive visualizations of segmented anatomical structures or lesions can help patients to better understand their condition. For instance, finding out a lung nodule or an abnormality in an endoscopy image offers patients a clear representation of their medical situation, aiding in informed discussions with healthcare providers and improving their engagement in their treatment plans.
>
> ## [bfV2-W2] Value of text prompts.
>
> While fully supervised methods have demonstrated high accuracy in medical image segmentation, text prompts provide distinct advantages:
> - **Clinical Utility:** Clinicians who may not have the radiological expertise to interpret complex imaging studies often need guidance on lesion locations. They may already have preliminary information about the presence of a lesion but unclear about its exact location (eg., pneumothorax, lump). Text prompts enable segmentation by helping clinicians locate lesions, e.g., identifying a lump in the lungs based on prior imaging.
> - **Multi-class Adaptability:** Traditional methods are constrained by pre-defined categories, limiting adaptability. Text-promptable segmentation offers the flexibility to dynamically define and target specific categories, enhancing clinical relevance.
> - **Performance Impact:** Our experiments demonstrate that text prompts contribute to segmentation accuracy. Without prompts, the average Dice score for five lesions decreases by 9.0% (from 72.69% to 63.69%), underscoring their importance.
>
> ## [bfV2-W3] Explanation of the box head.
>
> The mask head is implemented using dynamic convolution [1]. It takes multi-scale features from the feature pyramid network (FPN), concatenates them with relative coordinates, and uses a controller to generate convolutional parameters. **The relative coordinates are generated by the box head,** which  provides  a coarse location as an initial reference for mask prediction, as you correctly noted.
>
> ## [bfV2-W4] Comparison with traditional supervised models.
>
> We greatly value your suggestion that including comparisons with state-of-the-art models like nnU-Net strengthens the evaluation. Using the official nnU-Net code, we conducted experiments on abdominal organ segmentation (due to time constraints) and trained eight separate models, one for each organ, as nnU-Net specializes in per-organ segmentation. The average Dice score is 75.46%. Unlike nnU-Net, our model is universal, training on all organs and lesions simultaneously. Notably, our method outperforms Swin-UNet, achieving a 3.1% improvement in the average Dice score (76.48% vs. 73.38%).
>
> | Method | Aorta | Left Kidney | Right Kidney | Liver | Spleen | Stomach | Pancreas | Gallbladder | Average |
> | ----- | :-----: | :-----: | :-----: | :-----: | :-----: |  :-----: |  :-----: |  :-----: |  :-----: |
> | nn-UNet [2] | 92.17 | 79.59 | 78.42 | 87.56 | 81.18 | 68.07 | 56.84 | 59.87 | 75.46 |
> | Swin-UNet [3] | 77.85 | 82.34 | 75.60 | 90.07 | 86.97 | 66.89 | 52.49 | 54.84 | 73.38 |
> | Ours | 86.14 | 87.53 | 84.16 | 90.32 | 88.41 | 67.35 | 47.61 | 60.29 | 76.48 |

---

> ### Author Response · Authors · 2024-11-24
> **(2/2) Response to Reviewer bfV2**
>
> ## [bfV2-W5] Exploration of text prompts.
>
> We have expanded our experiments to evaluate the impact of various prompt types and designs:
>
> **1. different specificity** for anatomical entities. We tested simplified prompts containing only class names,  eg. "a MRI of the myocardium", "a CT of the liver tumor", "an ultrasound image of the prostate". These prompts yielded Dice scores of  75.65% for organs (-5.12%) and 67.61% for lesions (-5.08%), indicating that detailed and well-designed prompts are effective to enhance segmentation performance.
>
> **2. different attributes** for anatomical entities. We introduced new prompts incorporating attributes like [position/location], [boundary] and [density]. For example, for myocardium:
>
> - Position: "The myocardium is located between the endocardium and epicardium of the heart."
> - Boundary: "The boundaries of the myocardium on MRI are well-defined, showing a clear demarcation."
> - Density: "The myocardium typically exhibits low signal intensity on T2-weighted images."
>
> These prompts achieved Dice scores of 80.84% for organs (+0.07%) and 73.91% for lesions (+0.22%), demonstrating that well-designed text prompts effectively enhance segmentation accuracy.
>
> | Description version | Dice score (organ) | Dice score (lesion) |
> | ----- | :-----: | :-----: |
> | Different specificity | 75.65 | 67.61 |
> | Different attributes | 80.84 | 73.91 |
> | Original TPP | 80.77 | 72.69 |
>
>
> ## [bfV2-W6] Inconsistency of ambiguous expressions.
>
> Thank you for pointing out the ambiguity in terminology. We will carefully replace "the referred object" at P2 L92, P3 L146 and L159, P5 L243 and P6 with consistent and precise expressions.
>
> >[1] Tian, Zhi, Chunhua Shen, and Hao Chen. "Conditional convolutions for instance segmentation." Computer Vision-ECCV 2020: 16th European Conference, Glasgow, UK, August 23-28, 2020, Proceedings, Part I 16. Springer International Publishing, 2020.
>
> >[2] Isensee, Fabian, et al. "nnU-Net: a self-configuring method for deep learning-based biomedical image segmentation." Nature methods 18.2 (2021): 203-211.
>
> >[3] Cao, Hu, et al. "Swin-unet: Unet-like pure transformer for medical image segmentation." European conference on computer vision. Cham: Springer Nature Switzerland, 2022.

---

> > ### Comment · Reviewer_bfV2 · 2024-11-27
> > **Response to the rebuttal**
> >
> > Thank you for the rebuttal. After reading the feedback from other reviewers, I noticed that most of us share similar concerns about the incompleteness of the comparison experiments. In the revised version, the comparisons across several datasets still lack some important baselines in the medical domain, such as nnUNet, Swin UNETR, CLIP-driven Universal Model, and UniverSeg (ICCV). Additionally, the validation of the method's performance in 3D scenarios remains insufficient. Based on the current experimental setup, the proposed method's performance has not been validated on important 3D benchmarks, such as the MSD dataset. Furthermore, while the paper claims the superiority of the proposed method in few-shot/one-shot settings, it lacks comparisons with state-of-the-art methods in these settings.
> >
> > Overall, the experiments presented in the paper do not adequately support the claimed contributions. I am not yet convinced that the paper makes a substantial contribution to medical image segmentation, and therefore, I choose to maintain my original score. Perhaps the paper still needs a bit more work.

---

> > > ### Author Response · Authors · 2024-12-02
> > > **Response to Reviewer bfV2**
> > >
> > > Thank you very much for your additional feedback and for highlighting areas of improvement.
> > >
> > > We have conducted comparisons with strong baselines, including nn-UNet, Swin-UNet, and CLIP-driven Universal Model, on the BTCV dataset—a well-recognized 3D benchmark which contains 8 abdominal organs. The results are summarized below:
> > >
> > > | Method | Aorta | Left Kidney | Right Kidney | Liver | Spleen | Stomach | Pancreas | Gallbladder | Average |
> > > | ----- | :-----: | :-----: | :-----: | :-----: | :-----: |  :-----: |  :-----: |  :-----: |  :-----: |
> > > | CLIP-driven [1] | 88.31 | 84.94 | 81.04 | 93.03 | 79.83 | 65.88 | 42.74 | 53.85 | 73.70 |
> > > | nn-UNet [2] | 92.17 | 79.59 | 78.42 | 87.56 | 81.18 | 68.07 | 56.84 | 59.87 | 75.46 |
> > > | Swin-UNet [3] | 77.85 | 82.34 | 75.60 | 90.07 | 86.97 | 66.89 | 52.49 | 54.84 | 73.38 |
> > > | Ours | 86.14 | 87.53 | 84.16 | 90.32 | 88.41 | 67.35 | 47.61 | 60.29 | **76.48** |
> > >
> > > Our method achieves superior performance on this 3D benchmark, despite being a universal solution trained across 18 datasets, in contrast to [1]-[3], which are trained individually for each dataset.
> > >
> > > Regarding validation on other 3D benchmarks, such as the MSD dataset, we acknowledge this as a valuable suggestion. We also appreciate your note on few-shot/one-shot settings. While these experiments are currently limited due to time constraints, we plan to incorporate them in a revised version of the manuscript to further support the claimed contributions.

---

### Official Review · Reviewer_SfoW · 2024-11-03

**Soundness:** 4
**Presentation:** 3
**Contribution:** 3
**Rating:** 6
**Confidence:** 4

**Summary:**

The authors propose a new task: Referring Medical Image Sequence Segmentation, which aims to segment anatomical regions corresponding to given text prompts. To address this task, the authors propose a Text-Promptable Propagation (TPP) model, which takes sequential slices (either from 3D volume or videos) and text prompts as inputs and outputs the predicted masks. The authors have created a new dataset that consists of 18 3D/video medical datasets. Experiments were conducted against several referring video object segmentation algorithms, where the proposed method achieved better performance.

**Strengths:**

1. The paper is well-written and easy to follow.
2. The experiments are thorough and demonstrate clear advantage over other referring video object segmentation algorithms.
3. The proposed Triple Propagation that enforces temporal relationship between consecutive slices is novel.

**Weaknesses:**

1. The motivation for Referring Medical Image Sequence Segmentation remains a bit unclear to me. Specifically, (a) although the authors propose a unified framework for 2D and 3D segmentation task and claim this to be an advantage this setting, no evaluation is conducted on 2D datasets; (b) although the closed set label space is a limitation of traditional segmentation, I don't think this will be a severe problem if the closed set covers most important region of an input (for example, [1] includes 25 organs and 6 types of tumors that cover all common organs and tumor types, making the closed set almost the full set of the label space).
2. There is no ablation on the internal design of the TPP network, for example (a) how effective is the "Cross-modal Prompt Fusion" against a simple fusion strategy that averages or concatenates the image and textual features; (b) the prompt is repeated N_q times, meaning each image will have N_q outputs, and thus the computation cost is multiplied by N_q times as well. The analysis on the computation cost (such as FLOPS) is not included and the effect of N_q is not studied.

[1] CLIP-Driven Universal Model for Organ Segmentation and Tumor Detection. https://arxiv.org/pdf/2301.00785.

**Questions:**

My main concern is 1b. To address this concern, I'd like to see some comparison against 3D segmentation algorithms, such as [1] and [2]. The experiments can be conducted on a specific type of body locations (for example only training and evaluating on abdomen data) due to the limited time for rebuttal.


[1] CLIP-Driven Universal Model for Organ Segmentation and Tumor Detection. https://arxiv.org/pdf/2301.00785.
[2] nnU-Net: a self-configuring method for deep learning-based biomedical image segmentation. https://www.nature.com/articles/s41592-020-01008-z

---

> ### Author Response · Authors · 2024-11-24
> **Response to Reviewer SfoW**
>
> Thank you for your encouraging recognition of our work and constructive feedback. We have carefully addressed each of your points in detail and hope that these clarifications effectively resolve your concerns.
>
> ## [SfoW-W1(a)] Has any evaluation been conducted on 2D datasets?
>
> Medical image sequences include  both temporally related frames (e.g., in videos) and spatially related slices (e.g., in volumes). Our unified framework bridges 2D and 3D segmentation tasks, addressing diverse clinical needs. We classify datasets such as CT and MRI as 3D datasets, while **ultrasound and endoscopy images are categorized as 2D datasets**. Examples of 2D datasets include CAMUS, Micro-Ultrasound Prostate Segmentation Dataset, CVC-ClinicDB, CVC-ColonDB, ETIS, and ASU-Mayo.
>
> ## [SfoW-W1(b)] Comparison against 3D segmentation algorithms.
>
> The work *CLIP-driven universal model for organ segmentation and tumor detection* [1] is an excellent reference, which  we will cite in the revised paper. Using the official code and the same data splits as ours, we trained this model on abdominal organs for 200 epochs. It achieved an average Dice score of 73.70%, while our method attained 76.48%, demonstrating superior performance. Our method benefits from customized prompts, enabling domain-specific adaptability. Notably, our model is significantly more computationally efficient, with 130.77 GFLOPs compared to over 300 GFLOPs for [1]. For comparison with *nn-UNet* [2], we trained the model using the official codes of [2] under default settings, resulting in an average Dice score of 75.46%. The corresponding results are presented below:
> - Table 1: Comparison of computational cost.
>
>     | Method | FLOPs (G) |
>     | ----- | ----- |
>     | Clip-driven [1] | >300 |
>     | Ours (TPP) | 130 |
>
> - Table 2: Comparison against 3D segmentation algorithms on abdominal organs.
>
>     | Method | Aorta | Kidney (L) | Kidney (R) | Liver | Spleen | Stomach | Pancreas | Gallbladder | Average |
>     | ----- | :-----: | :-----: | :-----: | :-----: | :-----: |  :-----: |  :-----: |  :-----: |  :-----: |
>     | CLIP-driven [1] | 88.31 | 84.94 | 81.04 | 93.03 | 79.83 | 65.88 | 42.74 | 53.85 | 73.70 |
>     | nn-UNet [2] | 92.17 | 79.59 | 78.42 | 87.56 | 81.18 | 68.07 | 56.84 | 59.87 | 75.46 |
>     | Ours | 86.14 | 87.53 | 84.16 | 90.32 | 88.41 | 67.35 | 47.61 | 60.29 | 76.48 |
>
> ## [SfoW-W2(a)] Ablation study on fusion designs.
>
> Thank you for pointing out the need for ablation studies on fusion strategies. We have conducted additional experiments to evaluate the effectiveness of our proposed "Cross-modal Prompt Fusion" against simpler strategies. The results confirm that our "Cross-modal Prompt Fusion" significantly outperforms these  alternatives, demonstrating its efficacy in leveraging image and text features for segmentation.
>
> - Table 3: Ablation studies on fusion designs.
>
>     | Fusion design | Dice score (organ) | Dice score (lesion) |
>     | ----- | :-----: | :-----: |
>     | Average | 78.88 | 68.89 |
>     | Concatenation | 77.50 | 69.64 |
>     | Ours | **80.77** | **72.69** |
>
>
> ## [SfoW-W2(b)] Analysis of N_q and computation cost.
>
> As you correctly pointed out, the first image has N_q queries. Due to our propagation strategy, the best prediction of the first image is propagated to subsequent images, **reducing the number of queries to  just one for the rest of the images**. This optimization significantly lowers the overall computational burden.
> - Trainable params: 52.97M.
> - FLOPs: 130.77 GFLOPs, considerably lighter than [1] (FLOPs > 300G).
>
> The effect of N_q was studied under our propagation strategy. The results demonstrate that tracking the referred object becomes more robust when using a selection of queries (5->1->1), validating the design choice.
>
> - Table 4: Analysis on query selection. The first column represents the number of queries from Slice 1 to Slice 3.
>     | The number of queries for slices | Dice score (organ) | Dice score (lesion) |
>     | :-----: | :-----: | :-----: |
>     | 5 -> 5 -> 5| 79.47 | 70.98 |
>     | 5 -> 3 -> 1| 78.47 | 71.67 |
>     | 5 -> 1 -> 1| 80.77 | 72.69 |
>
>
> >[1] Liu, Jie, et al. "Clip-driven universal model for organ segmentation and tumor detection." Proceedings of the IEEE/CVF International Conference on Computer Vision. 2023.
>
> >[2] Isensee, Fabian, et al. "nnU-Net: a self-configuring method for deep learning-based biomedical image segmentation." Nature methods 18.2 (2021): 203-211.

---

> > ### Comment · Reviewer_SfoW · 2024-11-26
> >
> > Thank you for the response. Based on the newer result that the purposed method surpasses standard supervised 3D segmentation algorithms, I have raised my score to weak accept.

---

### Official Review · Reviewer_QiAh · 2024-11-04

**Soundness:** 3
**Presentation:** 3
**Contribution:** 3
**Rating:** 5
**Confidence:** 4

**Summary:**

This paper aims to solve the challenges of limited interaction between 2D and 3D segmentation models, in parallel of adding interactive prompt to provide human-guided context for segmentation in real clinical scenarios. This paper contribution can be summarized as follows:
1) Proposed an innovative task: referring medical image sequence segmentation
2) Proposed a baseline mode Text-Promptable Propagation (TPP) to exploit the intrinsic relationship among sequenctial images and thir associated textual descriptions
3) Benchmarked the model across 18 different datasets across 4 modalities.

**Strengths:**

The strength of this paper can be summarized as follows:
1) Performed experiments with 18 datasets
2) Compared baselines with SAM-2
3) Aimed to create new task for medical imaging domain

**Weaknesses:**

The weakness of this paper can be summarized as follows:
1) Really confused about the clinical scenarios or the medical problem that this paper is targeting
2) Limited experiments have been performance and haven't compared to the medical domain state-of-the-art
3) Insufficient clarity on the innovation of the proposed model, it seems like this model is a composition of so many current design blocks

**Questions:**

1) I am confused about the clinical problem that the paper are trying to solve, as 2D snapshots (e.g. CT) is possible to be demonstrated in the temporal domain (i.e. same subject but takes the image in different time), due to the needs of quick imaging in the clinical scenario. I am wondering if this is the problem that this paper are trying to solve?

As the medical image sequences you are referring in the paper is really similar to the consecutive slices of the same 3D image or video, it will be great to have more clarity on this.

2) Previous similar work have been demonstrated to adapt text-prompt across 9 modalities:
- Zhao, Theodore, et al. "BiomedParse: a biomedical foundation model for image parsing of everything everywhere all at once." arXiv preprint arXiv:2405.12971 (2024).

You can claim that your work is an extension idea from this, but I haven't seen any citation / experiments comparison with this. It will be great if you can add / use similar text scenario in BiomedParse to have a comparison.

3) In Table 4, as the experiment is performed with SAM-2, it should compare with the current SAM-based state-of-the-art model and even 2D completely supervised model (i.e. nnUNet, Swin-UNet), as the final goal is to enhance the segmentation performance for all slices input.
- Ma, Jun, et al. "Segment anything in medical images." Nature Communications 15.1 (2024): 654.
- Isensee, Fabian, et al. "nnU-Net: a self-configuring method for deep learning-based biomedical image segmentation." Nature methods 18.2 (2021): 203-211.
- Cao, Hu, et al. "Swin-unet: Unet-like pure transformer for medical image segmentation." European conference on computer vision. Cham: Springer Nature Switzerland, 2022.

4) As one the innovation on your model is to adapt descriptive text for enhancing the slice-by-slice relationship for segmentation, I also want to know the effectiveness of different versions of description, seems like there is no experiments to benchmark different versions of description, although you have provided the text prompt in the appendix. Wondering if this will be one of the core to affect segmentation performance.

5) Also, is your model generated binary segmentation and use text to refer the class semantics? Seems like you have used focal loss during training and I assume that the class label for the anatomy have been used. It will be great to see the performance if we don't need the class label for training and just adapt the text as a loss function.

---

> ### Author Response · Authors · 2024-11-24
> **(1/2) Response to Reviewer QiAh**
>
> Thank you for your encouraging recognition of our work and constructive feedback. We have carefully addressed each of your points in detail and hope these clarifications effectively resolve your concerns.
>
> ## [QiAh-W1, Q1] What clinical scenario or problem is the paper targeting?
>
> - **Definition of medical image sequences.** As you correctly pointed out, medical image sequences consist of consecutive slices of the same 3D image or video captured **at one inspection**. These include both temporally related frames from videos and spatially related slices within volumes. Such sequences differ from CT scans of the same patient taken at different time points. Modern medical imaging modalities are increasingly dominated by sequence-based data, such as CT or MRI slices, ultrasound scans, and endoscopy videos. Our work specifically targets medical image sequence segmentation—segmenting target objects within these sequences, guided by medical text prompts.
>
> - **Clinical scenarios**. In clinical practice, pre-defined text prompts are utilized to identify and segment anatomical structures or lesions that are challenging to recognize, such as pneumothorax. This text-promptable method significantly aids clinicians who may **lack radiological expertise** to interpret complex imaging studies. Radiologists often provide guidance to clinicians, such as **pointing out the exact location** of a pulmonary nodule. Our method automates this process, thereby saving time and improving diagnostic efficiency. By embedding this contextual understanding into our model, we aim to bridge the gap between radiological expertise and broader clinical applications.
>
> ## [QiAh-W2, Q3] Comparison with medical domain state-of-the-art.
>
> We greatly value your suggestion to strengthen the comparative experiments. In response, we have incorporated evaluations against state-of-the-art methods, including the SAM-based MedSAM [1] and supervised models such as Swin-UNet [2] and nnUNet [3].
>
> Due to time constraints, the experiments were conducted on the BTCV dataset, which includes 8 abdominal organs. We evaluate MedSAM under the inference tutorial with text-based prompts, which adopts the CLIP text model as the text encoder. Although MedSAM’s training dataset includes abdominal organs from the FLARE challenge, its average Dice score of 27.60% suggests that it struggles with zero-shot segmentation tasks. In contrast, our proposed method demonstrates superior generalization capabilities, as shown by the results presented in Table 5 of the main text.
>
> Results for the 2D supervised models (Swin-UNet and nnUNet) are included in the table below. Notably,  our method consistently outperforms state-of-the-art models in the medical domain.
>
> | Method | Dice score (average) |
> | ----- | :-----: |
> | MedSAM [1] | 27.60 |
> | Swin-UNet [2] | 73.38 |
> | nnUNet [3] | 75.46 |
> | Ours | **76.48** |
>
> ## [QiAh-Q2] Comparison with BiomedParse.
>
> BiomedParse [4] is a groundbreaking work in the field, and we will appropriately cite it in the revised manuscript. While BiomedParse demonstrates strong general-purpose capabilities across multiple modalities, our focus is on clinically relevant segmentation tasks driven by domain-specific text prompts. Our approach prioritizes automation and accuracy to address clinical needs, particularly in challenging scenarios where text-promptable segmentation aligns with the requirements of clinicians.

---

> ### Author Response · Authors · 2024-11-24
> **(2/2) Response to Reviewer QiAh**
>
> ## [QiAh-Q4] Experiments benchmarking different versions of descriptions.
> We conducted experiments with two types of prompt variations to evaluate their impact on segmentation performance:
>
> **1. different specificity for anatomical entities.**
> Simplified prompts with only class names resulted in Dice scores of 75.65% for organs (-5.12%) and 67.61% for lesions (-5.08%). Examples of such prompts include: "an MRI of the myocardium", "a CT of the liver tumor", "an ultrasound image of the prostate". The results demonstrate that detailed, descriptive prompts significantly enhance segmentation performance compared to simplified ones.
>
> **2. different attributes for anatomical entities.** We add new prompts describing anatomical entities with attributes such as [position/location], [boundary] and [density]. For example, for the myocardium:
> - Position: "The myocardium is located between the endocardium and epicardium of the heart."
> - Boundary: "The boundaries of the myocardium on MRI are well-defined, showing a clear demarcation."
> - Density: "The myocardium typically exhibits low signal intensity on T2-weighted images."
>
> These attribute-rich prompts yielded Dice scores of 80.84% for organs (+0.07%) and 73.91% for lesions (+0.22%), indicating that richer, attribute-based descriptions enhance segmentation accuracy when sufficient contextual information is provided.
>
> | Description version | Dice score (organ) | Dice score (lesion) |
> | ----- | :-----: | :-----: |
> | Different specificity | 75.65 | 67.61 |
> | Different attributes | 80.84 | 73.91 |
> | Original TPP | 80.77 | 72.69 |
>
>
> ## [QiAh-Q5] Does your model generate binary segmentation using text to refer to class semantics?
>
> Yes, we use text to refer to the class semantics. But in our model, the class label  is used solely to indicate **whether the current target is the referred object (binary 0/1)** during training. It does not contain semantic information about the target anatomy itself. We sincerely hope this explanation addresses your concern.
>
> >[1] Ma, Jun, et al. "Segment anything in medical images." Nature Communications 15.1 (2024): 654.
>
> >[2] Cao, Hu, et al. "Swin-unet: Unet-like pure transformer for medical image segmentation." European conference on computer vision. Cham: Springer Nature Switzerland, 2022.
>
> >[3] Isensee, Fabian, et al. "nnU-Net: a self-configuring method for deep learning-based biomedical image segmentation." Nature methods 18.2 (2021): 203-211.
>
> >[4] Zhao, Theodore, et al. "BiomedParse: a biomedical foundation model for image parsing of everything everywhere all at once." arXiv preprint arXiv:2405.12971 (2024).

---

### Meta-Review · Area_Chair_vCxX · 2024-12-22

**Metareview:**

This paper introduces a new task, Referring Medical Image Sequence Segmentation, which aims to segment anatomical regions in medical image sequences based on text prompts. To address this, the authors propose the Text-Promptable Propagation (TPP) model, leveraging cross-modal prompt fusion and a Transformer-based triple-propagation strategy to exploit spatial, temporal, and textual relationships for segmentation. The task and method are claimed to address challenges in integrating 2D and 3D segmentation models and enabling human-guided context in clinical scenarios. A comprehensive benchmark dataset, Ref-MISS, comprising 18 datasets across diverse imaging modalities, was curated to evaluate the method. Experiments demonstrated that TPP outperforms existing referring video object segmentation algorithms.

Strength: The paper's strengths lie in the new approach for Referring Medical Image Sequence Segmentation, using cross-modal prompt fusion and triple-propagation techniques to address the need for context-aware segmentation. Besides, it introduces a large-scale, diverse dataset covering 18 public datasets across 4 imaging modalities and 20 anatomical entities, providing a valuable resource for future research. The focus on text-guided prompts for medical image segmentation highlights its practical relevance for varying clinical scenarios.

Weakness: Most reviewers raise the concern that the experimental evaluation is limited, lacking comparisons with state-of-the-art methods in the medical domain and important baselines. The assumption that 3D imaging slices and video frames can be processed uniformly may be questionable: the paper does not adequately justify why it is beneficial to treat 3D volumes as sequential data instead of using direct 3D models, which may better capture spatial coherence. Moreover, the experiments do not sufficiently validate the claimed contributions, particularly in 3D scenarios. The added value of text prompts for segmentation is unclear, given the effectiveness of fully supervised methods.

Overall, considering the paper's contribution and the remaining concerns about the experiment evaluation and results comparison with regard to the claims, I suggest rejection and that the paper could be improved by a major revision.

**Additional Comments On Reviewer Discussion:**

The author provides a detailed response in the rebuttal including many additional experiment results, and most of the reviewers have responded and followed up quite a few rounds. From the reviewers' follow-up, the concerns about the experiment evaluation and results comparison still stand out. After reading the paper, review comments and rebuttals, I agree with these comments as described in the weakness section above.

---

### Decision · Program_Chairs · 2025-01-22

Reject